# Near-Optimal Reinforcement Learning in Dynamic Treatment Regimes

**Junzhe Zhang**
Department of Computer Science
Columbia University
New York, NY 10027
junzhez@cs.columbia.edu

**Elias Bareinboim**
Department of Computer Science
Columbia University
New York, NY 10027
eb@cs.columbia.edu

## Abstract

A dynamic treatment regime (DTR) consists of a sequence of decision rules, one per stage of intervention, that dictates how to determine the treatment assignment to patients based on evolving treatments and covariates' history. These regimes are particularly effective for managing chronic disorders and is arguably one of the key aspects towards more personalized decision-making. In this paper, we investigate the online reinforcement learning (RL) problem for selecting optimal DTRs provided that observational data is available. We develop the first adaptive algorithm that achieves near-optimal regret in DTRs in online settings, without any access to historical data. We further derive informative bounds on the system dynamics of the underlying DTR from confounded, observational data. Finally, we combine these results and develop a novel RL algorithm that efficiently learns the optimal DTR while leveraging the abundant, yet imperfect confounded observations.

## 1 Introduction

In medical practice, a patient typically has to be treated at multiple stages; the physician repeatedly adapts each treatment, tailored to the patient's time-varying, dynamic state (e.g., level of virus, results of diagnostic tests). Dynamic treatment regimes (DTRs) [18] provide an attractive framework of personalized treatments in longitudinal settings. Operationally, a DTR consists of decision rules that dictate what treatment to provide at each stage, given the patient's evolving conditions and history. These decision rules are alternatively known as adaptive treatment strategies [12, 13, 19, 33, 34] or treatment policies [16, 37, 38]. DTRs offer an effective vehicle for personalized management of chronic conditions, including cancer, diabetes, and mental illnesses [36].

Consider the DTR instance regarding the treatment of alcohol dependence [19, 6], which is graphically represented in Fig. 1a . Based on the condition of alcohol dependant patients ($S_1$), the physician may prescribe a medication or behavioral therapy ($X_1$). Patients are classified as responders or non-responders ($S_2$) based on their level of heavy drinking within the next two months. The physician then must decide whether to continue the initial treatment or switch to an augmented plan combining both medication and behavioral therapy ($X_2$). The unobserved covariate $U$ summarizes all the unknown factors about the patient. We are interested in the primary outcome $Y$ that is the percentage of abstinent days over a 12-month period. The treatment policy $\pi$ in this set-up is a sequence of decision rules $x_1 \leftarrow \pi_1(s_1), x_2 \leftarrow \pi_2(s_1, s_2, x_1)$ selecting the values of $X_1, X_2$ based on the history.

Policy learning in a DTR setting is concerned with finding an optimal policy $\pi$ that maximizes the primary outcome $Y$. The main challenge is that since the parameters of the DTR are often unknown, it's not immediate how to directly compute the consequences of executing the policy $do(\pi)$, i.e., the expected value $E_\pi[Y]$. Most of the current work in the causal inference literature focus on trying to identify this quantity, $E_\pi[Y]$, from finite observational data and causal assumptions about the data-

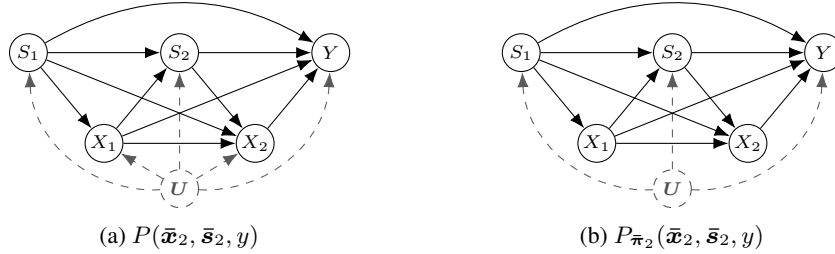

(a) $P(\bar{\boldsymbol{x}}_2, \bar{\boldsymbol{s}}_2, y)$            (b) $P_{\bar{\boldsymbol{\pi}}_2}(\bar{\boldsymbol{x}}_2, \bar{\boldsymbol{s}}_2, y)$

Figure 1: Causal diagrams of (a) a DTR with $K = 2$ stages of intervention; and (b) a DTR in (a) under sequential interventions $do(X_1 \sim \pi_1(X_1|S_1), X_2 \sim \pi_2(X_2|S_1, S_2, X_1))$.

generating mechanisms (commonly through causal graphs and potential outcomes). Several criteria and algorithms have been developed [23, 28, 4]. For instance, a criterion called *sequential backdoor* [24] permits one to determine whether causal effects can be obtained by covariate adjustment. This condition is also referred to as *conditional ignorability* or unconfoundedness [27, 18]: there exists no *unobserved confounders* (UCs) that simultaneously affects the treatment at any stage and all the subsequent outcomes given a set of observed covariates. Whenever ignorability holds, a number of efficient estimation procedures exist, including popular methods based on the propensity score [26], inverse probability of treatment weighting [21, 25], and Q-learning [31, 20].

In general, the combination of observational data and causal assumptions does not always lead to point-identification [23, Ch. 3-4]. An alternative is to randomize patients' treatments at each stage based on the previous decisions and observed outcomes; for instance, one popular strategy is known as the sequential multiple assignment randomized trail (SMART) [19]. By the virtue of randomization, the sequential backdoor condition is entailed. However, in practice, performing a randomized experiment in the actual environment can be extremely costly and undesirable (due to unintended consequences), especially for domains where humans are the main research subjects (e.g., medicine, epidemiology, and psychology). Reinforcement learning (RL) [31] provides a unique opportunity to efficiently learning DTRs due to its nature of balancing exploration and exploitation. A typical RL agent learns by conducting adaptive, sequential experimentation: it repeatedly adjusts the policy that is currently deployed based on the past outcomes. The goal is to learn an optimal policy while minimizing the experimental cost. Efficient RL algorithms have been successfully developed to very general settings such as Markov decision processes (MDPs) [30, 11, 32], where a finite state is statistically sufficient to summarize the treatments and covariates' history. Variations of this setting include multi-armed bandits [1], partially-observable MDP [10, 2], and factored MDPs [22].

Our focus here is on learning a policy for an unknown DTR while leveraging the observational data. This is a challenging setting for both causal inference and RL. As an example, consider data collected from an unknown behavior policy of the DTR in Fig. 1a (i..e, $x_1 \leftarrow f_1(s_1, u), x_2 \leftarrow f_2(s_1, s_2, x_1, u)$, where both $U$ and $\{f_1, f_2\}$ are unobserved), which is materialized in the form of the observational distribution $P(x_1, x_2, s_1, s_2, y)$ [23, pp. 205]. The existence of the unmeasured confounder $U$ leads to an immediate violation of the sequential backdoor criterion (e.g., due to the spurious path $X_1 \leftarrow U \rightarrow Y$), which implies that the effect of the policy $E_{\boldsymbol{\pi}}[Y]$ is not identifiable [23, Ch. 4.4]. On the other hand, existing RL algorithms are not applicable either, which can be seen by noting that DTRs are inherently non-Markovian – in words, the initial treatment $X_1$ directly affects the outcome $Y$. Even though an heuristic approach may be pursued (e.g., Thompson Sampling [35]), and could eventually converge, the same is still not optimal since it's oblivious to all the observational data. [1]. Indeed, it is acknowledged in the literature [7, 8] that the "development of statistically sound estimation and inference techniques" for online RL settings "seem to be another very important research direction", especially when the increasing use of mobiles devices allows for the possibility of continuous monitoring and just-in-time intervention.

The goal of this paper is to overcome these challenges. We will introduce novel RL strategies capable of optimizing an unknown DTR while efficiently leveraging the imperfect, but large amounts of observational data. In particular, our contributions are as follows: (1) We introduce the first algorithm (UC-DTR (Alg. 1)) that reaches the near-optimal regret bound in the pure DTR setting, without

observational data; (2) We derive novel bounds capable of exploiting observational data based on the DTR structure (Thms. 5 and 6), which are provably tight; (3) We develop a novel algorithm (UC$^c$-DTR (Alg. 2)) that efficiently incorporates these bounds and accelerates learning in the online setting. Our results are validated on randomly generated DTRs and multi-stage clinical trials on cancer treatment.

## 1.1 Preliminaries

In this section, we introduce the basic notation and definitions used throughout the paper. We use capital letters to denote variables ($X$) and small letters for their values ($x$). Let $\mathcal{X}$ represent the domain of $X$ and $|\mathcal{X}|$ its dimension. We consistently use the abbreviation $P(x)$ to represent the probabilities $P(X = x)$. $\bar{\boldsymbol{X}}_k$ stands for a sequence $\{X_1, \ldots, X_k\}$ ($\emptyset$ if $k < 1$), and $\bar{\boldsymbol{\mathcal{X}}}_k$ represents its domain, i.e., $\mathcal{X}_1 \times \cdots \times \mathcal{X}_k$. Further, we denote by $I_{\{\cdot\}}$ the indicator function.

The basic semantical framework of our analysis rests on *structural causal models* (SCM) [23, Ch. 7]. A SCM $M$ is a tuple $\langle \boldsymbol{U}, \boldsymbol{V}, \boldsymbol{F}, P(\boldsymbol{u}) \rangle$ where $\boldsymbol{U}$ is a set of exogenous (unobserved) variables and $\boldsymbol{V}$ is a set of endogenous (observed) variables. $\boldsymbol{F}$ is a set of structural functions where $f_i \in \boldsymbol{F}$ decides the values of $V_i \in \boldsymbol{V}$ taking as argument a combination of other endogenous and exogenous variables (i.e., $V_i \leftarrow f_i(\boldsymbol{PA}_i, \boldsymbol{U}_i), \boldsymbol{PA}_i \subseteq \boldsymbol{V}, \boldsymbol{U}_i \subseteq \boldsymbol{U}$). The values of $\boldsymbol{U}$ are drawn from the distribution $P(\boldsymbol{u})$, and induce an observational distribution $P(\boldsymbol{v})$ [23, pp. 205]. Each SCM is associated with a causal diagram in the form of a directed acyclic graph $G$, where nodes represent endogenous variables, dashed nodes exogenous variables, and arrows stand for functional relations (e.g., see Fig. 1).

An intervention on a set of endogenous variables $\boldsymbol{X}$, denoted by $do(\boldsymbol{x})$, is an operation where values of $\boldsymbol{X}$ are set to constants $\boldsymbol{x}$, regardless of how they were ordinarily determined (through the functions $\{f_X : \forall X \in \boldsymbol{X}\}$). For a SCM $M$, let $M_{\boldsymbol{x}}$ be a sub-model of $M$ induced by intervention $do(\boldsymbol{x})$. The interventional distribution $P_{\boldsymbol{x}}(\boldsymbol{y})$ induced by $do(\boldsymbol{x})$ is the distribution over variables $\boldsymbol{Y}$ in the sub-model $M_{\boldsymbol{x}}$. For a more detailed discussion of SCMs, we refer readers to [23, Ch. 7].

## 2 Optimizing Dynamic Treatment Regimes

In this section, we will formalize the problem of online optimization in DTRs with confounded observations and provide an efficient solution. We start by defining DTRs in the structural semantics.

**Definition 1** (Dynamic Treatment Regime [18]). A dynamic treatment regime (DTR) is a SCM $\langle \boldsymbol{U}, \boldsymbol{V}, \boldsymbol{F}, P(\boldsymbol{u}) \rangle$ where the endogenous variables $\boldsymbol{V} = \{\bar{\boldsymbol{X}}_K, \bar{\boldsymbol{S}}_K, Y\}$; $K \in \mathbb{N}^+$ is the total stages of interventions. For stage $k = 1, \ldots, K$: (1) $X_k$ is a finite decision decided by a behavior policy $x_k \leftarrow f_k(\bar{\boldsymbol{s}}_k, \bar{\boldsymbol{x}}_{k-1}, \boldsymbol{u})$; (2) $S_k$ is a finite state decided by a transition function $s_k \leftarrow \tau_k(\bar{\boldsymbol{x}}_{k-1}, \bar{\boldsymbol{s}}_{k-1}, \boldsymbol{u})$. $Y$ is the primary outcome at the final state $K$, decided by a reward function $y \leftarrow r(\bar{\boldsymbol{x}}_K, \bar{\boldsymbol{s}}_K, \boldsymbol{u})$ bounded in $[0, 1]$. Values of exogenous variables $\boldsymbol{U}$ are drawn from the distribution $P(\boldsymbol{u})$.

A DTR $M^*$ induces an observational distribution $P(\bar{\boldsymbol{x}}_K, \bar{\boldsymbol{s}}_K, y)$. Fig. 1a shows the causal diagram of a DTR with $K = 2$ stages of interventions. A policy $\boldsymbol{\pi}$ for a DTR is a sequence of decision rules $\bar{\boldsymbol{\pi}}_K$, where each $\pi_k(x_k | \bar{\boldsymbol{s}}_k, \bar{\boldsymbol{x}}_{k-1})$ is a function mapping from the domain of histories $\bar{\boldsymbol{S}}_k, \bar{\boldsymbol{X}}_{k-1}$ up to stage $k$ to a distribution over decision $X_k$. A policy is called *deterministic* if the above mappings are from histories $\bar{\boldsymbol{S}}_k, \bar{\boldsymbol{X}}_{k-1}$ to the domain of decision $X_k$, i.e., $x_k \leftarrow \pi_k(\bar{\boldsymbol{s}}_k, \bar{\boldsymbol{x}}_{k-1})$. The collection of possible policies, depending on the domains of the history and decision, define a policy space $\boldsymbol{\Pi}$.

A policy $\boldsymbol{\pi}$ defines a sequence of stochastic interventions $do(X_1 \sim \pi_1(X_1 | \bar{\boldsymbol{S}}_1), \ldots, X_K \sim \pi_K(X_K | \bar{\boldsymbol{S}}_K, \bar{\boldsymbol{X}}_{K-1}))$, which induce an interventional distribution over variables $\bar{\boldsymbol{X}}_K, \bar{\boldsymbol{S}}_K, Y$, i.e.:

$$P_{\boldsymbol{\pi}}(\bar{\boldsymbol{x}}_K, \bar{\boldsymbol{s}}_K, y) = P_{\bar{\boldsymbol{x}}_K}(y | \bar{\boldsymbol{s}}_K) \prod_{k=0}^{K-1} P_{\bar{\boldsymbol{x}}_k}(s_{k+1} | \bar{\boldsymbol{s}}_k) \pi_{k+1}(x_{k+1} | \bar{\boldsymbol{s}}_{k+1}, \bar{\boldsymbol{x}}_k), \qquad (1)$$

where $P_{\bar{\boldsymbol{x}}_k}(s_{k+1} | \bar{\boldsymbol{s}}_k)$ is the transition distribution at stage $k$ and $P_{\bar{\boldsymbol{x}}_K}(y | \bar{\boldsymbol{s}}_K)$ is the reward distribution over the primary outcome. Fig. 1b describes a DTR under $K = 2$ stages of interventions $do(X_2 \sim \pi_1(X_1 | S_1), X_2 \sim \pi_2(X_2 | S_1, S_2, X_1))$. The expected cumulative reward of a policy $\boldsymbol{\pi}$ in a DTR $M^*$ is given by $V_{\boldsymbol{\pi}}(M^*) = E_{\boldsymbol{\pi}}[Y]$. We are searching for an optimal policy $\boldsymbol{\pi}^*$ that maximizes the cumulative reward, i.e., $\boldsymbol{\pi}^* = \arg\max_{\boldsymbol{\pi} \in \boldsymbol{\Pi}} V_{\boldsymbol{\pi}}(M^*)$. It is a well-known fact in decision theory that no stochastic policy can improve on the utility of the best deterministic policy (see, e.g., [15, Lem. 2.1]). Thus, in what follows, we will usually consider the policy space $\boldsymbol{\Pi}$ to be deterministic.

---
**Algorithm 1:** UC-DTR
---
**Input:** failure tolerance $\delta \in (0, 1)$.

1: **for all** episodes $t = 1, 2, \ldots$ **do**
2:     Define event counts $N^t(\bar{\boldsymbol{s}}_k, \bar{\boldsymbol{x}}_k)$ and $N^t(\bar{\boldsymbol{s}}_k, \bar{\boldsymbol{x}}_{k-1})$ for horizon $k = 1, \ldots, K$ prior to episode $t$ as, respectively, $\sum_{i=1}^{t-1} I_{\bar{\boldsymbol{S}}_k^i = \bar{\boldsymbol{s}}_k, \bar{\boldsymbol{X}}_k^i = \bar{\boldsymbol{x}}_k}$ and $\sum_{i=1}^{t-1} I_{\bar{\boldsymbol{S}}_k^i = \bar{\boldsymbol{s}}_k, \bar{\boldsymbol{X}}_{k-1}^i = \bar{\boldsymbol{x}}_{k-1}}$. Further, define reward counts $R^t(\bar{\boldsymbol{s}}_K, \bar{\boldsymbol{x}}_K)$ prior to episode $t$ as $\sum_{i=1}^{t-1} Y^i I_{\bar{\boldsymbol{S}}_K^i = \bar{\boldsymbol{s}}_K, \bar{\boldsymbol{X}}_K^i = \bar{\boldsymbol{x}}_K}$.
3:     Compute estimates $\hat{P}_{\bar{\boldsymbol{x}}_k}^t(s_{k+1}|\bar{\boldsymbol{s}}_k)$ and $\hat{E}_{\bar{\boldsymbol{x}}_K}^t[Y|\bar{\boldsymbol{s}}_K]$ as

$$\hat{P}_{\bar{\boldsymbol{x}}_k}^t(s_{k+1}|\bar{\boldsymbol{s}}_k) = \frac{N^t(\bar{\boldsymbol{s}}_{k+1}, \bar{\boldsymbol{x}}_k)}{\max\{1, N^t(\bar{\boldsymbol{s}}_k, \bar{\boldsymbol{x}}_k)\}}, \qquad \hat{E}_{\bar{\boldsymbol{x}}_K}^t[Y|\bar{\boldsymbol{s}}_K] = \frac{R^t(\bar{\boldsymbol{s}}_K, \bar{\boldsymbol{x}}_K)}{\max\{1, N^t(\bar{\boldsymbol{s}}_k, \bar{\boldsymbol{x}}_k)\}}.$$

4:     Let $\mathcal{M}_t$ denote a set of DTRs such that for any $M \in \mathcal{M}_t$, its transition probabilities $P_{\bar{\boldsymbol{x}}_k}(s_{k+1}|\bar{\boldsymbol{s}}_k)$ and reward $E_{\bar{\boldsymbol{x}}_K}[Y|\bar{\boldsymbol{s}}_K]$ are close to estimates $\hat{P}_{\bar{\boldsymbol{x}}_k}^t(s_{k+1}|\bar{\boldsymbol{s}}_k), \hat{E}_{\bar{\boldsymbol{x}}_K}^t[Y|\bar{\boldsymbol{s}}_K]$, i.e.,

$$\left\| P_{\bar{\boldsymbol{x}}_k}(\cdot|\bar{\boldsymbol{s}}_k) - \hat{P}_{\bar{\boldsymbol{x}}_k}^t(\cdot|\bar{\boldsymbol{s}}_k) \right\|_1 \le \sqrt{\frac{6|\mathcal{S}_{k+1}|\log(2K|\bar{\boldsymbol{\mathcal{S}}}_k||\bar{\boldsymbol{\mathcal{X}}}_k|t/\delta)}{\max\{1, N^t(\bar{\boldsymbol{s}}_k, \bar{\boldsymbol{x}}_k)\}}}, \tag{2}$$

$$\left| E_{\bar{\boldsymbol{x}}_K}[Y|\bar{\boldsymbol{s}}_K] - \hat{E}_{\bar{\boldsymbol{x}}_K}^t[Y|\bar{\boldsymbol{s}}_K] \right| \le \sqrt{\frac{2\log(2K|\boldsymbol{\mathcal{S}}||\boldsymbol{\mathcal{X}}|t/\delta)}{\max\{1, N^t(\bar{\boldsymbol{s}}_K, \bar{\boldsymbol{x}}_K)\}}}. \tag{3}$$

5:     Find the optimal policy $\boldsymbol{\pi}_t$ of an optimistic DTR $M_t \in \mathcal{M}_t$ such that

$$V_{\boldsymbol{\pi}_t}(M_t) = \max_{\boldsymbol{\pi} \in \boldsymbol{\Pi}, M \in \mathcal{M}_t} V_{\boldsymbol{\pi}}(M) \tag{4}$$

6:     Execute policy $\boldsymbol{\pi}_t$ for episode $t$ and observe the samples $\bar{\boldsymbol{S}}_K^t, \bar{\boldsymbol{X}}_K^t, Y^t$.
7: **end for**

---

Our goal is to optimize an unknown DTR $M^*$ based solely on the domains $\boldsymbol{\mathcal{S}} = \bar{\boldsymbol{\mathcal{S}}}_K$, $\boldsymbol{\mathcal{X}} = \bar{\boldsymbol{\mathcal{X}}}_K$ and the observational distribution $P(\bar{\boldsymbol{x}}_K, \bar{\boldsymbol{s}}_K, y)$ (i.e., both $\boldsymbol{F}, P(\boldsymbol{u})$ are unknown). The agent (e.g., a physician) learns through repeated experiments of episodes $t = 1, \ldots, T$. Each episode $t$ contains a complete DTR process: at stage $k$, the agent observes the state $S_k^t$, performs an intervention $do(X_k^t)$ and moves to the state $S_{k+1}^t$; the primary outcome $Y^t$ is received at the final stage $K$. The cumulative regret up to episode $T$ is defined as $R(T) = \sum_{t=1}^T (V_{\boldsymbol{\pi}^*}(M^*) - Y^t)$, i.e, the loss due to the fact that the agent does not always pick the optimal policy $\boldsymbol{\pi}^*$. We will assess and compare algorithms in terms of their regret $R(T)$. A desirable asymptotic property is to have $\lim_{T \to \infty} E[R(T)]/T = 0$, meaning that the agent eventually converges and finds the optimal policy $\boldsymbol{\pi}^*$.

## 2.1 The UC-DTR Algorithm

We now introduce a new RL algorithm for optimizing an unknown DTR, which we call UC-DTR. We will later prove that UC-DTR achieves near-optimal bound on the total regret given only the knowledge of the domains $\boldsymbol{\mathcal{S}}$ and $\boldsymbol{\mathcal{X}}$. Like many other online RL algorithms [1, 11, 22], UC-DTR follows the principle of *optimism under uncertainty* to balance exploration and exploitation. The algorithm generally works in phases of model learning, optimistic planning, and strategy execution.

The details of UC-DTR procedure can be found in Alg. 1. The algorithm proceeds in episodes and computes a new strategy $\boldsymbol{\pi}_t$ from samples $\{\bar{\boldsymbol{S}}_K^i, \bar{\boldsymbol{X}}_K^i, Y^i\}_{i=1}^{t-1}$ collected so far at the beginning of each episode $t$. Specifically, UC-DTR computes in Steps 1-3, the empirical estimates $\hat{E}_{\bar{\boldsymbol{x}}_K}^t[Y|\bar{\boldsymbol{s}}_K]$ of the expected reward $E_{\bar{\boldsymbol{x}}_K}[Y|\bar{\boldsymbol{s}}_K]$, and $\hat{P}_{\bar{\boldsymbol{x}}_k}^t(s_{k+1}|\bar{\boldsymbol{s}}_K)$ of the transitional probabilities $P_{\bar{\boldsymbol{x}}_k}(s_{k+1}|\bar{\boldsymbol{s}}_k)$ from experimental samples collected prior to episode $t$. In Step 4, a set $\mathcal{M}_t$ of plausible DTRs is defined in terms of confidence region around the the empirical estimates $\hat{E}_{\bar{\boldsymbol{x}}_K}^t[Y|\bar{\boldsymbol{s}}_K]$ and $\hat{P}_{\bar{\boldsymbol{x}}_k}^t(s_{k+1}|\bar{\boldsymbol{s}}_k)$. This guarantees that the true DTR $M^*$ is in the set $\mathcal{M}_t$ with high probability. In Step 5, UC-DTR computes the optimal policy $\boldsymbol{\pi}_t$ of the most optimistic instance $M_t$ in the family of DTRs $\mathcal{M}_t$ that induces the maximal optimal expected reward. This policy $\boldsymbol{\pi}_t$ is executed throughout episode $t$ and new samples $\bar{\boldsymbol{S}}_K^t, \bar{\boldsymbol{X}}_K^t, Y^t$ are collected (Step 6).

**Finding Optimistic DTRs**   The Step 5 of UC-DTR tries to find an optimal policy $\boldsymbol{\pi}_t$ for an optimistic DTR $M_t$. While the Bellman equation [5] allows one to optimize a fixed DTR, we need to find a DTR $M_t$ that gives the maximal optimal reward among all plausible DTRs in $\mathcal{M}_t$ given by Eq. (3).

We now introduce a method that extends standard dynamic programming planners [5] to solve this problem. We first combine all DTRs in $\mathcal{M}_t$ to get an extended DTR $M_+$. That is, we consider a DTR $M_+$ with continuous decision space $\bar{\boldsymbol{\mathcal{X}}}^+ = \bar{\boldsymbol{\mathcal{X}}}_K^+$, where for each horizon $k$, each action $\bar{\boldsymbol{x}}_k \in \bar{\boldsymbol{\mathcal{X}}}_k$, each admissible transition probabilities $P_{\bar{\boldsymbol{x}}_k}(s_{k+1}|\bar{\boldsymbol{s}}_k)$ according to Eq. (2), there is an action in $\bar{\boldsymbol{\mathcal{X}}}_k^+$ inducing the same probabilities $P_{\bar{\boldsymbol{x}}_k}(s_{k+1}|\bar{\boldsymbol{s}}_k)$. Similar arguments also apply to the expected reward $E_{\bar{\boldsymbol{x}}_K}[Y|\bar{\boldsymbol{s}}_K]$. Then, for each policy $\boldsymbol{\pi}_+$ on $M_+$, there is an DTR $M_t \in \mathcal{M}_t$ and a policy $\boldsymbol{\pi}_t \in \boldsymbol{\Pi}$ such that policies $\boldsymbol{\pi}_+$ and $\boldsymbol{\pi}_t$ induces the same transition probabilities on the respective DTR, and vice versa. Thus, solving the optimization problem in Eq. (4) is equivalent to finding an optimal policy $\boldsymbol{\pi}_+^*$ on the extended DTR $M_+$. Let $V^*(\bar{\boldsymbol{s}}_k, \bar{\boldsymbol{x}}_{k-1})$ denote the optimal value $E_{\boldsymbol{\pi}_+^*}[Y|\bar{\boldsymbol{s}}_k, \bar{\boldsymbol{x}}_{k-1}]$ in $M_+$. The Bellman equation on $M_+$ for $k = 1, \ldots, K-1$ is defined as follows:

$$V^*(\bar{\boldsymbol{s}}_k, \bar{\boldsymbol{x}}_{k-1}) = \max_{x_k} \left\{ \max_{P_{\bar{\boldsymbol{x}}_k}(\cdot|\bar{\boldsymbol{s}}_k) \in \mathcal{P}_k} \left\{ \sum_{s_{k+1}} V^*(\bar{\boldsymbol{s}}_{k+1}, \bar{\boldsymbol{x}}_k) P_{\bar{\boldsymbol{x}}_k}(s_{k+1}|\bar{\boldsymbol{s}}_k) \right\} \right\},$$

$$\text{and} \quad V^*(\bar{\boldsymbol{s}}_K, \bar{\boldsymbol{x}}_{K-1}) = \max_{x_K} \max_{E_{\bar{\boldsymbol{x}}_K}[Y|\bar{\boldsymbol{s}}_K] \in \mathcal{R}} E_{\bar{\boldsymbol{x}}_K}[Y|\bar{\boldsymbol{s}}_K], \tag{5}$$

where $\mathcal{R}$ and $\mathcal{P}_k$ are the convex polytope of parameters $E_{\bar{\boldsymbol{x}}_K}[Y|\bar{\boldsymbol{s}}_K]$ and $P_{\bar{\boldsymbol{x}}_k}(s_{k+1}|\bar{\boldsymbol{s}}_k)$ defined in Eqs. (2) and (3), respectively. The inner maximum in Eq. (5) is a linear program (LP) over the convex polytope $\mathcal{P}_k$ (or $\mathcal{R}$), which is solvable using standard LP algorithms.

## 2.2   Theoretical Analysis

We now analyze the asymptotic behavior of UC-DTR that will lead to a better understanding of its theoretical guarantees. Given space constraints, all proofs are provided in the full technical report [40, Appendix I]. The following proposition shows that the cumulative regret of UC-DTR after $T$ steps is at most $\tilde{\mathcal{O}}(K\sqrt{|\boldsymbol{\mathcal{S}}||\boldsymbol{\mathcal{X}}|T})^2$.

**Theorem 1.** *Fix a $\delta \in (0, 1)$. With probability (w.p.) of at least $1 - \delta$, it holds for any $T > 1$, the regret of UC-DTR with parameter $\delta$ is bounded by*

$$R(T) \leq 12K\sqrt{|\boldsymbol{\mathcal{S}}||\boldsymbol{\mathcal{X}}|T \log(2K|\boldsymbol{\mathcal{S}}||\boldsymbol{\mathcal{X}}|T/\delta)} + 4K\sqrt{T \log(2T/\delta)}. \tag{6}$$

It is also possible to obtain the instance-dependent bound on the expected regret. Let $\boldsymbol{\Pi}^-$ denote a set of sub-optimal policies $\{\boldsymbol{\pi} \in \boldsymbol{\Pi} : V_{\boldsymbol{\pi}}(M^*) < V_{\boldsymbol{\pi}^*}(M^*)\}$. For any $\boldsymbol{\pi} \in \boldsymbol{\Pi}^-$, let its gap in expected reward between the optimal policy $\boldsymbol{\pi}^*$ be $\Delta_{\boldsymbol{\pi}} = V_{\boldsymbol{\pi}^*}(M^*) - V_{\boldsymbol{\pi}}(M^*)$. We next derive the gap-dependent logarithmic bound on the expected regret of UC-DTR after $T$ steps.

**Theorem 2.** *For any $T \geq 1$, with parameter $\delta = \frac{1}{T}$, the expected regret of UC-DTR is bounded by*

$$E[R(T)] \leq \max_{\boldsymbol{\pi} \in \boldsymbol{\Pi}^-} \left\{ \frac{33^2 K^2 |\boldsymbol{\mathcal{S}}||\boldsymbol{\mathcal{X}}| \log(T)}{\Delta_{\boldsymbol{\pi}}} + \frac{32}{\Delta_{\boldsymbol{\pi}}^3} + \frac{4}{\Delta_{\boldsymbol{\pi}}} \right\} + 1. \tag{7}$$

Since Eq. (7) is a decreasing function relative to the gap $\Delta_{\boldsymbol{\pi}}$, the maximum of the regret in Thm. 2 is achieved with the second best policy $\boldsymbol{\pi}^- = \arg\min_{\boldsymbol{\pi} \in \boldsymbol{\Pi}^-} \Delta_{\boldsymbol{\pi}}$. We also provide a corresponding lower bound on the expected regret of any experimental algorithm.

**Theorem 3.** *For any algorithm $\mathcal{A}$, any natural numbers $K \geq 1$, and $|\boldsymbol{\mathcal{S}}^k| \geq 2, |\boldsymbol{\mathcal{X}}^k| \geq 2$ for any $k \in \{1, \ldots, K\}$, there is a DTR $M$ with horizon $K$, state domains $\boldsymbol{\mathcal{S}}$ and action domains $\boldsymbol{\mathcal{X}}$, such that the expected regret of $\mathcal{A}$ after $T \geq |\boldsymbol{\mathcal{S}}||\boldsymbol{\mathcal{X}}|$ episodes is as least*

$$E[R(T)] \geq 0.05\sqrt{|\boldsymbol{\mathcal{S}}||\boldsymbol{\mathcal{X}}|T} \tag{8}$$

Thm. 3 implies that for any DTR instance, the cumulative regret of $\Omega(\sqrt{|\boldsymbol{\mathcal{S}}||\boldsymbol{\mathcal{X}}|T})$ is inevitable. The regret upper bound $\tilde{\mathcal{O}}(K\sqrt{|\boldsymbol{\mathcal{S}}||\boldsymbol{\mathcal{X}}|T})$ in Thm. 1 is close to the lower bound $\Omega(\sqrt{|\boldsymbol{\mathcal{S}}||\boldsymbol{\mathcal{X}}|T})$ in Thm. 3, which means that UC-DTR is near-optimal provided with only the domains of state $\boldsymbol{\mathcal{S}}$ and actions $\boldsymbol{\mathcal{X}}$.

# 3 Learning from Confounded Observations

The results presented so far (Thms. 1 to 3) establish the dimension of the state-action domain $|\mathcal{S}||\mathcal{X}|$ as the an important parameter for the information complexity of online learning in DTRs. When domains $\mathcal{S} \times \mathcal{X}$ are high-dimensional, the cumulative regret will be significant for any online algorithm, no matter how sophisticated it might be. This observation suggests that we should explore other reasonable assumptions to address the issues of high-dimensional domains.

A natural approach is to utilize the abundant observational data, which could be obtained by passively observing other agents behaving in the environment. Despite all its power, the UC-DTR algorithm does not make use of any knowledge in the the observational distribution $P(\bar{s}_K, \bar{x}_K, y)$. For the remainder of this paper, we will present and study an efficient procedure to incorporate observational samples of $P(\bar{s}_K, \bar{x}_K, y)$, so that the performance of online learners could be improved.

When states $\bar{S}_K$ satisfy the *sequential backdoor* criterion [24] with respect to treatments $\bar{X}_K$ and the primary outcome $Y$, one could identify the transition probabilities $P_{\bar{x}_k}(s_{k+1}|\bar{s}_k)$ and expected reward $E_{\bar{x}_K}[Y|\bar{s}_k]$ from $P(\bar{s}_K, \bar{x}_K, y)$. The optimal policy is thus solvable using the standard off-policy learning methods such as Q-learning [31, 20]. However, issues of non-identifiability arise in the general settings where the sequential backdoor does not hold (e.g., see Fig. 1a).

**Theorem 4.** *Given $P(\bar{s}_K, \bar{x}_K, y) > 0$, there exists DTRs $M_1, M_2$ such that $P^{M_1}(\bar{s}_K, \bar{x}_K, y) = P^{M_2}(\bar{s}_K, \bar{x}_K, y) = P(\bar{s}_K, \bar{x}_K, y)$ while $P^{M_1}_{\bar{x}_K}(\bar{s}_K, y) \neq P^{M_2}_{\bar{x}_K}(\bar{s}_K, y)$.*

Thm. 4 is stronger than the standard non-identifiability results (e.g., [14, Thm. 1]). It shows that given *any* observational distribution $P(\bar{s}_K, \bar{x}_K, y)$, one to construct two DTRs both compatible with $P(\bar{s}_K, \bar{x}_K, y)$, but disagrees in the interventional probabilities $P_{\bar{x}_K}(\bar{s}_K, y)$.

## 3.1 Bounds and Partial Identification in DTRs

In this section, we consider a partial identification task in DTRs which bounds parameters of $P_{\bar{x}_k}(s_{k+1}|\bar{s}_k)$ and $E_{\bar{x}_K}[Y|\bar{s}_k]$ from the observational distribution $P(\bar{s}_K, \bar{x}_K, y)$. Our first result shows that the gap between causal quantities $P_{\bar{x}_k}(\bar{s}_{k+1})$ and $P_{\bar{x}_k}(\bar{s}_k)$ in a DTR is bounded by the gap between the corresponding observational distributions $P(\bar{s}_{k+1}, \bar{x}_k)$ and $P(\bar{s}_k, \bar{x}_k)$.

**Lemma 1.** *For a DTR, given $P(\bar{s}_K, \bar{x}_K, y)$, for any $k = 1, \dots, K - 1$,*

$$P_{\bar{x}_k}(\bar{s}_{k+1}) - P_{\bar{x}_k}(\bar{s}_k) \leq P(\bar{s}_{k+1}, \bar{x}_k) - P(\bar{s}_k, \bar{x}_k). \tag{9}$$

Lem. 1 allows one to derive informative bounds of transition probabilities $P_{\bar{x}_k}(s_{k+1}|\bar{s}_k)$ in a DTR, which are consistently estimable from the observational data $P(\bar{s}_K, \bar{x}_K)$.

**Theorem 5.** *For a DTR, given $P(\bar{s}_K, \bar{x}_K, y) > 0$, for any $k = 1, \dots, K - 1$,*

$$\frac{P(\bar{s}_{k+1}, \bar{x}_k)}{\Gamma(\bar{s}_k, \bar{x}_{k-1})} \leq P_{\bar{x}_k}(s_{k+1}|\bar{s}_k) \leq \frac{\Gamma(\bar{s}_{k+1}, \bar{x}_k)}{\Gamma(\bar{s}_k, \bar{x}_{k-1})}, \tag{10}$$

*where $\Gamma(\bar{s}_{k+1}, \bar{x}_k) = P(\bar{s}_{k+1}, \bar{x}_k) - P(\bar{s}_k, \bar{x}_k) + \Gamma(\bar{s}_k, \bar{x}_{k-1})$ and $\Gamma(s_1) = P(s_1)$.*

Bounds in Thm. 5 exploit the sequential functional relationships among states and treatments in the underlying DTR, which improve over the best-known bounds reported in [17, 3, 39]. Let $[a_{\bar{x}_k, \bar{s}_k}(s_{k+1}), b_{\bar{x}_k, \bar{s}_k}(s_{k+1})]$ denote the bound over $P_{\bar{x}_k}(s_{k+1}|\bar{s}_k)$ given by Eq. (10). We next show that $P_{\bar{x}_k}(s_{k+1}|\bar{s}_k) \in [a_{\bar{x}_k, \bar{s}_k}(s_{k+1}), b_{\bar{x}_k, \bar{s}_k}(s_{k+1})]$ is indeed optimal without additional assumption.

**Theorem 6.** *Given $P(\bar{s}_K, \bar{x}_K, y) > 0$, for any $k \in \{1, \dots, K - 1\}$, there exists DTRs $M_1, M_2$ such that $P^{M_1}(\bar{s}_K, \bar{x}_K, y) = P^{M_2}(\bar{s}_K, \bar{x}_K, y) = P(\bar{s}_K, \bar{x}_K, y)$ while $P^{M_1}_{\bar{x}_k}(s_{k+1}|\bar{s}_k) = a_{\bar{x}_k, \bar{s}_k}(s_{k+1})$, $P^{M_2}_{\bar{x}_k}(s_{k+1}|\bar{s}_k) = b_{\bar{x}_k, \bar{s}_k}(s_{k+1})$.*

Thm. 6 ensures the optimality of Thm. 5. Suppose there exists a bound $[a'_{\bar{x}_k, \bar{s}_k}(s_{k+1}), b'_{\bar{x}_k, \bar{s}_k}(s_{k+1})]$ strictly contained in $[a_{\bar{x}_k, \bar{s}_k}(s_{k+1}), b_{\bar{x}_k, \bar{s}_k}(s_{k+1})]$. By Thm. 6, we could always find DTRs $M_1, M_2$ that are compatible with the observational data $P(\bar{s}_K, \bar{x}_K, y)$ while their transition probabilities $P_{\bar{x}_k}(s_{k+1}|\bar{s}_k)$ lie outside of the bound $[a'_{\bar{x}_k, \bar{s}_k}(s_{k+1}), b'_{\bar{x}_k, \bar{s}_k}(s_{k+1})]$, which is a contradiction.

As a corollary, one could apply methods of Lem. 1 and Thm. 5 to bound expected rewards $E_{\bar{x}_K}[Y|\bar{s}_k]$ from $P(\bar{s}_K, \bar{x}_K, y)$. The optimality of the derived bounds follows immediately after Thm. 6.

---

**Algorithm 2:** Causal UC-DTR (UC$^c$-DTR)

---

    **Input:** failure tolerance $\delta \in (0, 1)$, causal bounds $\mathcal{C}$.

1: Let $\mathcal{M}^c$ denote a set of DTRs compatible with causal bounds $\mathcal{C}$, i.e., for any $M \in \mathcal{M}^c$, its causal quantities $P_{\bar{\boldsymbol{x}}_k}(s_{k+1}|\bar{\boldsymbol{s}}_k)$ and $E_{\bar{\boldsymbol{x}}_K}[Y|\bar{\boldsymbol{s}}_K]$ satisfy Eq. (13) and Eq. (14) respectively.
2: **for all** episodes $t = 1, 2, \ldots$ **do**
3:    Execute Steps 2-4 of UC-DTR (Alg. 1).
4:    Find the optimal policy $\boldsymbol{\pi}_t$ of an optimistic DTR $M_t$ in $\mathcal{M}_t^c = \mathcal{M}_t \cap \mathcal{M}^c$ such that

$$V_{\boldsymbol{\pi}_t}(M_t) = \max_{\boldsymbol{\pi} \in \boldsymbol{\Pi}, M \in \mathcal{M}_t^c} V_{\boldsymbol{\pi}}(M) \tag{12}$$

5:    Execute policy $\boldsymbol{\pi}_t$ for episode $t$ and observe the samples $\bar{\boldsymbol{S}}_K^t, \bar{\boldsymbol{X}}_K^t, Y^t$.
6: **end for**

---

**Corollary 1.** *For a DTR, given $P(\bar{\boldsymbol{s}}_K, \bar{\boldsymbol{x}}_K, y) > 0$,*

$$\frac{E[Y|\bar{\boldsymbol{s}}_K, \bar{\boldsymbol{x}}_K]P(\bar{\boldsymbol{s}}_K, \bar{\boldsymbol{x}}_K)}{\Gamma(\bar{\boldsymbol{s}}_K, \bar{\boldsymbol{x}}_{K-1})} \le E_{\bar{\boldsymbol{x}}_K}[Y|\bar{\boldsymbol{s}}_k] \le 1 - \frac{(1 - E[Y|\bar{\boldsymbol{s}}_K, \bar{\boldsymbol{x}}_K])P(\bar{\boldsymbol{s}}_K, \bar{\boldsymbol{x}}_K)}{\Gamma(\bar{\boldsymbol{s}}_K, \bar{\boldsymbol{x}}_{K-1})}. \tag{11}$$

Since $E[Y|\bar{\boldsymbol{s}}_K, \bar{\boldsymbol{x}}_K] \in [0, 1]$, the bounds in Eq. (11) are contained in $[0, 1]$ and are thus informative. The bounds developed so far are functions of the observational distribution $P(\bar{\boldsymbol{s}}_K, \bar{\boldsymbol{x}}_K, y)$ which is identifiable by the sampling process, and so generally can be estimated consistently. Specifically, we estimate the bounds in Thm. 5 and Corol. 1 by the corresponding sample mean estimates. Standard results of large-deviation theory are thus applicable to control the uncertainties due to finite samples.

### 3.2 The Causal UC-DTR Algorithm

Our goal in this section is to introduce a simple, yet principled approach for leveraging the new-found bounds defined in Thm. 5 and Corol. 1, hopefully improving the performance of UC-DTR procedure.

For $k = 1, \ldots, K - 1$, let $\mathcal{C}_k$ denote a set of bounds over transition probabilities $P_{\bar{\boldsymbol{x}}_k}(s_{k+1}|\bar{\boldsymbol{s}}_k)$, i.e.,

$$\mathcal{C}_k = \left\{ \forall \bar{\boldsymbol{s}}_{k+1}, \bar{\boldsymbol{x}}_k : P_{\bar{\boldsymbol{x}}_k}(s_{k+1}|\bar{\boldsymbol{s}}_k) \in \left[ a_{\bar{\boldsymbol{x}}_k, \bar{\boldsymbol{s}}_k}(s_{k+1}), b_{\bar{\boldsymbol{x}}_k, \bar{\boldsymbol{s}}_k}(s_{k+1}) \right] \right\}. \tag{13}$$

Similarly, let $\mathcal{C}_K$ denote a set of bounds over the conditional expected reward $E_{\bar{\boldsymbol{x}}_K}[Y|\bar{\boldsymbol{s}}_K]$, i.e.,

$$\mathcal{C}_K = \left\{ \forall \bar{\boldsymbol{s}}_K, \bar{\boldsymbol{x}}_K : E_{\bar{\boldsymbol{x}}_K}[Y|\bar{\boldsymbol{s}}_K] \in \left[ a_{\bar{\boldsymbol{x}}_K, \bar{\boldsymbol{s}}_K}, b_{\bar{\boldsymbol{x}}_K, \bar{\boldsymbol{s}}_K} \right] \right\}. \tag{14}$$

We denote by $\mathcal{C}$ a set of bounds $\{\mathcal{C}_1, \ldots, \mathcal{C}_K\}$ on the system dynamics of the DTR, called *causal bounds*. Our procedure Causal UC-DTR (for short, UC$^c$-DTR) is summarized in Alg. 2. UC$^c$-DTR is similar to the original UC-DTR but exploits causal bounds $\mathcal{C}$. It maintains a set of possible DTRs $\mathcal{M}^c$ compatible with the causal bounds $\mathcal{C}$ (Step 1). Before each episode $t$, it computes the optimal policy $\boldsymbol{\pi}_t$ of an optimistic DTRs $M_t$ in set $\mathcal{M}_t^c = \mathcal{M}_t \cap \mathcal{M}^c$ (Step 3). Similar to UC-DTR, $\boldsymbol{\pi}_t$ could be obtained by solving LPs defined in Eq. (5) subject to additional causal constraints Eqs. (13) and (14).

We next analyze asymptotic properties of UC$^c$-DTR, showing that it consistently outperforms UC-DTR. Let $\left\| \mathcal{C}_k \right\|_1$ denote the maximal L1 norm of any parameter in $\mathcal{C}_k$, i.e., for any $k = 1, \ldots, K - 1$,

$$\left\| \mathcal{C}_k \right\|_1 = \max_{\bar{\boldsymbol{x}}_k, \bar{\boldsymbol{s}}_k} \sum_{s_{k+1}} \left| a_{\bar{\boldsymbol{x}}_k, \bar{\boldsymbol{s}}_k}(s_{k+1}) - b_{\bar{\boldsymbol{x}}_k, \bar{\boldsymbol{s}}_k}(s_{k+1}) \right|, \quad \text{and} \quad \left\| \mathcal{C}_K \right\|_1 = \max_{\bar{\boldsymbol{x}}_K, \bar{\boldsymbol{s}}_K} \left| a_{\bar{\boldsymbol{x}}_K, \bar{\boldsymbol{s}}_K} - b_{\bar{\boldsymbol{x}}_K, \bar{\boldsymbol{s}}_K} \right|.$$

Further, let $\left\| \mathcal{C} \right\|_1 = \sum_{k=1}^K \left\| \mathcal{C}_k \right\|_1$. The total regret of UC$^c$-DTR after $T$ steps is bounded as follows.

**Theorem 7.** *Fix a $\delta \in (0, 1)$. With probability of at least $1 - \delta$, it holds for any $T > 1$, the regret of UC$^c$-DTR with parameter $\delta$ and causal bounds $\mathcal{C}$ is bounded by*

$$R(T) \le \min \left\{ 12K \sqrt{|\mathcal{S}||\mathcal{X}|T \log(2K|\mathcal{S}||\mathcal{X}|T/\delta)}, \left\| \mathcal{C} \right\|_1 T \right\} + 4K \sqrt{T \log(2T/\delta)}. \tag{15}$$

It is immediate from Thm. 7 that the regret bound in Eq. (15) is smaller than the bound given by Eq. (6) if $T < 12^2 |\mathcal{S}||\mathcal{X}| \log(2K|\mathcal{S}||\mathcal{X}|T/\delta)/\left\| \mathcal{C} \right\|_1^2$. This means that UC$^c$-DTR has a head start over UC-DTR when the causal bounds $\mathcal{C}$ are informative, i.e., the dimension $\left\| \mathcal{C} \right\|_1$ is small.

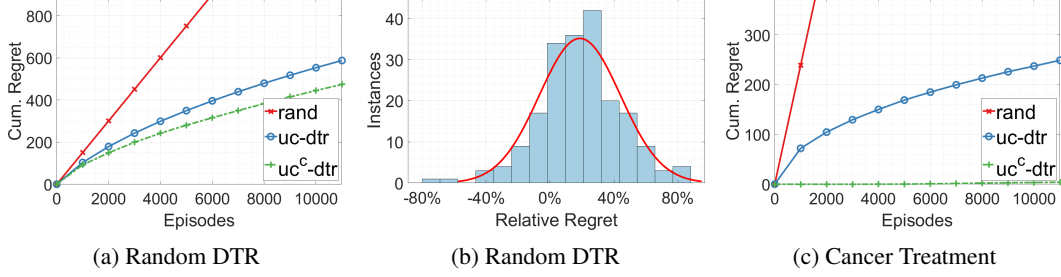

(a) Random DTR       (b) Random DTR       (c) Cancer Treatment

Figure 2: Simulations comparing online learners that are randomized (*rand*), adaptive (*uc-dtr*) and causally enhanced (*$uc^c$-dtr*). Graphs are rendered in high resolution and can be zoomed in.

We could also witness the improvements of causal bounds on the total expected regret. Let $\mathbf{\Pi}_C^-$ be the set of sub-optimal policies that their maximal expected rewards over instances in $\mathcal{M}^c$ are no less than the true optimal value $V_{\pi^*}(M^*)$, i.e., $\mathbf{\Pi}_C^- = \{\pi \in \mathbf{\Pi}^- : \max_{M \in \mathcal{M}^c} V_\pi(M) \geq V_{\pi^*}(M^*)\}$. The following is the instance-dependent bound on the total regret of UC$^c$–DTR after $T$ steps.

**Theorem 8.** *For any $T \geq 1$, with parameter $\delta = \frac{1}{T}$ and causal bounds $\mathcal{C}$, the expected regret of* UC$^c$-DTR *is bounded by*

$$E[R(T)] \leq \max_{\pi \in \mathbf{\Pi}_C^-} \left\{ \frac{33^2 K^2 |\mathcal{S}||\mathcal{X}| \log(T)}{\Delta_\pi} + \frac{32}{\Delta_\pi^3} + \frac{4}{\Delta_\pi} \right\} + 1. \tag{16}$$

Since $\mathbf{\Pi}_C^- \subseteq \mathbf{\Pi}^-$, it follows that the regret bound in Thm. 8 is small than or equal to Eq. (7), i.e., UC$^c$–DTR consistently dominates UC–DTR in terms of the performance. For instance, in a multi-armed bandit model (i.e., 1-stage DTR with $S_1 = \emptyset$) with optimal reward $\mu^*$, the regret of UC$^c$–DTR is $\mathcal{O}(|\mathcal{X}| \log(T)/\Delta_x)$ where $\Delta_x$ is the smallest gap among sub-optimal arms $x$ satisfying $b_x \geq \mu^*$.

## 4 Experiments

We demonstrate our algorithms on several dynamic treatment regimes, including randomly generated DTRs, and the survival model in the context of multi-stage cancer treatment. We found that our algorithms could efficiently found the optimal policy; the observational data typically improve the convergence rate of online RL learners despite the confounding bias.

In all experiments, we test sequentially randomized trials (*rand*), UC–DTR algorithm (*uc-dtr*) and the causal UC–DTR (*$uc^c$-dtr*) with causal bounds derived from $1 \times 10^5$ confounded observational samples. Each experiment lasts for $T = 1.1 \times 10^4$ episodes. The parameter $\delta = \frac{1}{KT}$ for *uc-dtr* and *$uc^c$-dtr* where $K$ is the total stages of interventions. For all algorithms, we measure their cumulative regret over 200 repetitions. We refer readers to the complete technical report [40, Appendix II] for the more details on the experimental set-up.

**Random DTRs** We generate 200 random instances and observational distributions of the DTR described in Fig. 1. We assume treatments $X_1, X_2$, states $S_1, S_2$ and primary outcome $Y$ are all binary variables; values of each variable are decided by their corresponding unobserved counterfacutals $S_{2_{x_1}}, X_{2_{x_1}}, Y_{\bar{x}_2}$ following definitions in [3, 9]. The probabilities of the joint distribution $P(s_1, x_1, s_{2_{x_1}}, x_{2_{x_1}}, y_{\bar{x}_2})$ are drawn randomly over $[0, 1]$. The cumulative regrets average among all random DTRs are reported in Fig. 2a. We find that online methods (*uc-dtr*, *$uc^c$-dtr*) dominate randomized assignments (*rand*); RL learners that leverage causal bounds (*$uc^c$-dtr*) consistently dominates learners that do not (*uc-dtr*). Fig. 2b reports the relative improvement in total regrets of *$uc^c$-dtr* compared to *uc-dtr* among 200 instances: *$uc^c$-dtr* outperforms *uc-dtr* in over $80\%$ of generated DTRs. This suggests that causal bounds derived from the observational data are beneficial in most instances.

**Cancer Treatment** We test the survival model of the two-stage clinical trial conducted by the Cancer and Leukemia Group B [16, 37]. Protocol 8923 was a double-blind, placebo controlled two-stage trial reported by [29] examining the effects of infusions of granulocyte-macrophage colony-stimulating factor (GM-CSF) after initial chemotherapy in patients with acute myelogenous

leukemia (AML). Standard chemotherapy for AML could place patients at increased risk of death due to infection or bleeding-related complications. GM-CSF administered after chemotherapy might assist patient recovery, thus reducing the number of deaths due to such complications. Patients were randomized initially to GM-CSF or placebo following standard chemotherapy. Later, patients meeting the criteria of complete remission and consenting to further participation were offered a second randomization to one of two intensification treatments.

Fig. 1a describes the DTR of this two-stage trail. $X_1$ represents the initial GM-CSF administration and $X_2$ represents the intensification treatment; the initial state $S_1 = \emptyset$ and $S_2$ indicates the complete remission after the first treatment; the primary outcome $Y$ indicates the survival of patients at the time of recording. We generate observational samples using *age* of patients as UCs $\boldsymbol{U}$. The cumulative regrets average among all random DTRs are reported in Fig. 2b. We find that *rand* performs worst among all strategies; *uc-dtr* finds the optimal policy with sub-linear regrets. Interestingly, *uc$^c$-dtr* converges almost immediately, suggesting that causal bounds derived from confounded observations could significantly improve the performance of online learners.

## 5   Conclusion

In this paper, we investigated the online reinforcement learning problem for selecting the optimal DTR provided with abundant, yet imperfect observations made about the underlying environment. We first presented an online RL algorithm with near-optimal regret bounds in DTRs solely based on the knowledge about state-action domains. We further derived causal bounds about the system dynamics in DTRs from the observational data. These bounds could be incorporated in a simple, yet principled way to improve the performance of online RL learners. In today's healthcare, for example, the growing use of mobile devices opens new opportunities in continuous monitoring of patients' conditions and just-in-time interventions. We believe that our results constitute a significant step towards the development of a more principled and robust science of precision medicine.

**Acknowledgments**

This research is supported in parts by grants from IBM Research, Adobe Research, NSF IIS-1704352, and IIS-1750807 (CAREER).

## Footnotes

[1]Standard off-policy RL methods such as Q-Learning rely on the condition of sequential backdoor, thus not applicable for the confounded observational data. For a more elaborate discussion, see [7, Ch. 3.5]

[2]$\tilde{\mathcal{O}}(\cdot)$ is similar to $\mathcal{O}(\cdot)$ but ignores log-terms, i.e., $f = \tilde{\mathcal{O}}(g)$ if and only if $\exists k$, $f = \mathcal{O}(g \log^k(g))$.

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
