[Supplementary Material]

# "Near-Optimal Reinforcement Learning in Dynamic Treatment Regimes" Supplemental Material

## 1 Appendix I. Proofs

2 In this section, we provide proofs for the theoretical results presented in the main text.

### 3 Proof of Theorems 1 to 3

4 We start by introducing necessary notations for the proof. We say an episode $t$ is $\epsilon$-bad if $V_{\boldsymbol{\pi}^*}(M^*) -$
5 $Y^t \geq \epsilon$. Let $T_\epsilon$ be the number of episodes taken by UC-DTR that are $\epsilon$-bad. Let $L_\epsilon$ denote the indices
6 of the $\epsilon$-bad episodes up to episode $T$. The cumulative regret $R_\epsilon(T)$ in $\epsilon$-bad episodes up to episode
7 $T$ is defined as $R_\epsilon(T) = \sum_{t \in L_\epsilon} V_{\boldsymbol{\pi}^*}(M^*) - Y^t$. For any $k = 1, \ldots, K$, we define event counts
8 $N(\bar{\boldsymbol{s}}_k, \bar{\boldsymbol{x}}_k)$ in total episodes $T$ as $N(\bar{\boldsymbol{s}}_k, \bar{\boldsymbol{x}}_k) = \sum_{t=1}^T I_{\bar{\boldsymbol{S}}_k^t = \bar{\boldsymbol{s}}_k, \bar{\boldsymbol{X}}_k^t = \bar{\boldsymbol{x}}_k}$. Finally, we denote by $\boldsymbol{\mathcal{H}}^t$ the
9 history up to episode $t$, i.e., $\boldsymbol{\mathcal{H}}^t = \{\bar{\boldsymbol{X}}_K^1, \bar{\boldsymbol{S}}_K^1, Y^1, \ldots, \bar{\boldsymbol{X}}_K^t, \bar{\boldsymbol{S}}_K^t, Y^t\}$.

10 **Lemma 2.** *Fix $\delta \in (0,1)$, with probability at least $1 - \delta$,*

$$\sum_{t \in L_\epsilon} \left( E_{\bar{\boldsymbol{X}}_K^t}[Y|\bar{\boldsymbol{S}}_K^t] - Y^t \right) \leq \sqrt{\frac{T_\epsilon \log(1/\delta)}{2}}.$$

11 *Proof.* Let $\boldsymbol{D}^T$ denote the sequence $\{\bar{\boldsymbol{X}}_K^1, \bar{\boldsymbol{S}}_K^1, \ldots, \bar{\boldsymbol{X}}_K^T, \bar{\boldsymbol{S}}_K^T\}$. Rewards $Y^t$ are independent vari-
12 ables by conditioning on $\boldsymbol{D}^T = \boldsymbol{d}^T$. Applying Hoeffding's inequality gives:

$$P\left( \sum_{t \in L_\epsilon} \left( E_{\bar{\boldsymbol{x}}_K^t}[Y|\bar{\boldsymbol{s}}_K^t] - Y^t \right) \geq \sqrt{\frac{T_\epsilon \log(1/\delta)}{2}} \mid \boldsymbol{d}^T \right) \leq \delta.$$

13 We thus have:

$$P\left( \sum_{t \in L_\epsilon} \left( E_{\bar{\boldsymbol{X}}_K^t}[Y|\bar{\boldsymbol{S}}_K^t] - Y^t \right) \geq \sqrt{\frac{T_\epsilon \log(1/\delta)}{2}} \mid \boldsymbol{d}^T \right) \leq \delta \sum_{\boldsymbol{d}^T} P(\boldsymbol{d}^T) = \delta. \qquad \square$$

14 **Lemma 3.** *Fix $\epsilon > 0$, $\delta \in (0,1)$. With probability (w.p.) of at least $1 - \delta$, it holds for any $T > 1$,*
15 *$R_\epsilon(T)$ of UC-DTR with parameter $\delta$ is bounded by*

$$R_\epsilon(T) \leq 12K\sqrt{|\boldsymbol{\mathcal{S}}||\boldsymbol{\mathcal{X}}|T_\epsilon \log(2K|\boldsymbol{\mathcal{S}}||\boldsymbol{\mathcal{X}}|T/\delta)} + 4K\sqrt{T_\epsilon \log(2T/\delta)}$$

16 *Proof.* Let $M^*$ denote the underlying DTR. Recall that $\mathcal{M}_t$ is a set of DTR instances such that for
17 any $M \in \mathcal{M}_t$, its system dynamics satisfy

$$\left\| P_{\bar{\boldsymbol{x}}_k}^M(\cdot|\bar{\boldsymbol{s}}_k) - \hat{P}_{\bar{\boldsymbol{x}}_k}^t(\cdot|\bar{\boldsymbol{s}}_k) \right\|_1 \leq \sqrt{\frac{6|\mathcal{S}_{k+1}|\log(2K|\bar{\boldsymbol{\mathcal{S}}}_k||\bar{\boldsymbol{\mathcal{X}}}_k|t/\delta)}{\max\{1, N^t(\bar{\boldsymbol{s}}_k, \bar{\boldsymbol{x}}_k)\}}}, \tag{16}$$

$$\left| E_{\bar{\boldsymbol{x}}_K}^M[Y|\bar{\boldsymbol{s}}_K] - \hat{E}_{\bar{\boldsymbol{x}}_K}^t[Y|\bar{\boldsymbol{s}}_K] \right| \leq \sqrt{\frac{2\log(2K|\boldsymbol{\mathcal{S}}||\boldsymbol{\mathcal{X}}|t/\delta)}{\max\{1, N^t(\bar{\boldsymbol{s}}_K, \bar{\boldsymbol{x}}_K)\}}}. \tag{17}$$

18 By union bounds and Hoeffding's inequality (following a similar argument in [4, C.1]),

$$P(M^* \in \mathcal{M}_t) \leq \frac{\delta}{4t^2}.$$

19 Since $\sum_{t=1}^{\infty} \frac{1}{4t^2} \leq \frac{\pi^2}{24}\delta < \frac{\delta}{2}$, it follows that with probability at least $1 - \frac{\delta}{2}$, $M^* \in \mathcal{M}^t$ for all episodes
20 $t = 1, 2, \ldots$.

21 For the remainder of the proof, we will assume that $M^* \in \mathcal{M}_t$ for all $t$. Let $E_{\bar{\boldsymbol{x}}_K}^{M_t}[Y|\bar{\boldsymbol{s}}_K]$ denote the
22 conditional expected reward in the optimistic DTR $M_t$. We can write $R_\epsilon(T)$ as:

$$R_\epsilon(T) = \sum_{t \in L_\epsilon} \left( V_{\boldsymbol{\pi}^*}(M^*) - E_{\bar{\boldsymbol{X}}_K^t}^{M_t}[Y|\bar{\boldsymbol{S}}_K^t] \right) \tag{18}$$

$$+ \sum_{t \in L_\epsilon} \left( E_{\bar{\boldsymbol{X}}_K^t}^{M_t}[Y|\bar{\boldsymbol{S}}_K^t] - E_{\bar{\boldsymbol{X}}_K^t}[Y|\bar{\boldsymbol{S}}_K^t] \right) \tag{19}$$

$$+ \sum_{t \in L_\epsilon} \left( E_{\bar{\boldsymbol{X}}_K^t}[Y|\bar{\boldsymbol{S}}_K^t] - Y^t \right). \tag{20}$$

23 We will next derive bounds over $R_\epsilon(T)$ by bounding quantities in Eqs. (18) to (20) separately.

24 **Bounding Eq. (18)** For any DTR $M$ and policy $\boldsymbol{\pi}$, let $V_{\boldsymbol{\pi}}(\bar{\boldsymbol{s}}_k, \bar{\boldsymbol{x}}_{k-1}; M) = E_{\boldsymbol{\pi}}^M[Y|\bar{\boldsymbol{s}}_k, \bar{\boldsymbol{x}}_{k-1}]$ and
25 $V_{\boldsymbol{\pi}}(\bar{\boldsymbol{s}}_k, \bar{\boldsymbol{x}}_k; M) = E_{\boldsymbol{\pi}}^M[Y|\bar{\boldsymbol{s}}_k, \bar{\boldsymbol{x}}_k]$. Since $M^* \in \mathcal{M}_t$, we must have $V_{\boldsymbol{\pi}^*}(s_1; M^*) \leq V_{\boldsymbol{\pi}_t}(s_1; M_t)$,
26 i.e., the maximal expected reward of the optimal reward in the optimistic $M_t$ is no less than that in
27 the underlying DTR $M^*$ for any initial state $s_1$. Further, since $\boldsymbol{\pi}_t$ is deterministic, for any stage $k$
28 and DTR $M$,

$$V_{\boldsymbol{\pi}_t}(\bar{\boldsymbol{S}}_k^t, \bar{\boldsymbol{X}}_{k-1}^t; M) = V_{\boldsymbol{\pi}_t}(\bar{\boldsymbol{S}}_k^t, \bar{\boldsymbol{X}}_{k-1}^t; M). \tag{21}$$

29 We thus have

$$V_{\boldsymbol{\pi}^*}(M^*) - E_{\bar{\boldsymbol{X}}_K^t}^{M_t}[Y|\bar{\boldsymbol{S}}_K^t] \leq V_{\boldsymbol{\pi}^*}(M^*) - V_{\boldsymbol{\pi}^*}(\bar{\boldsymbol{S}}_1^t; M^*) + V_{\boldsymbol{\pi}_t}(\bar{\boldsymbol{S}}_1^t, \bar{\boldsymbol{X}}_1^t; M^*) - E_{\bar{\boldsymbol{X}}_K^t}^{M_t}[Y|\bar{\boldsymbol{S}}_K^t].$$

30 Let $M_t(k)$ denote a combined DTR obtained from $M^*$ and $M_t$ such that

31       • for $i = 0, 1, \ldots, k-1$, its transition probability $P_{\bar{\boldsymbol{x}}_i}^{M_t(k)}(s_{i+1}|\bar{\boldsymbol{s}}_i)$ coincides with the transi-
32       tion probability $P_{\bar{\boldsymbol{x}}_i}(s_{i+1}|\bar{\boldsymbol{s}}_i)$ in the real DTR $M^*$;

33       • for $i = k, \ldots, K-1$, its transition probability $P_{\bar{\boldsymbol{x}}_i}^{M_t(k)}(s_{i+1}|\bar{\boldsymbol{s}}_i)$ coincides with the transition
34       probability $P_{\bar{\boldsymbol{x}}_i}^{M_t}(s_{i+1}|\bar{\boldsymbol{s}}_i)$ in the optimistic $M_t$

35 This is, for any $\boldsymbol{\pi} \in \boldsymbol{\Pi}$, the interventional distribution $P_{\boldsymbol{\pi}}^{M_t(k)}(\bar{\boldsymbol{x}}_K, \bar{\boldsymbol{s}}_K, y)$ factorizes as follows:

$$P_{\boldsymbol{\pi}}^{M_t(k)}(\bar{\boldsymbol{x}}_K, \bar{\boldsymbol{s}}_K, y) = P_{\bar{\boldsymbol{x}}_K}^{M_t}(y|\bar{\boldsymbol{s}}_K) \prod_{i=0}^{k-1} P_{\bar{\boldsymbol{x}}_i}(s_{i+1}|\bar{\boldsymbol{s}}_i)$$
$$\cdot \prod_{j=k}^{K-1} P_{\bar{\boldsymbol{x}}_j}^{M_t}(s_{i+1}|\bar{\boldsymbol{s}}_j) \prod_{l=1}^{K-1} \pi_{l+1}(x_{l+1}|\bar{\boldsymbol{s}}_{l+1}, \bar{\boldsymbol{x}}_l). \tag{22}$$

36 Obviously, $E_{\bar{\boldsymbol{X}}_K^t}^{M_t}[Y|\bar{\boldsymbol{S}}_K^t] = V_{\boldsymbol{\pi}_t}(\bar{\boldsymbol{S}}_K^t, \bar{\boldsymbol{X}}_K^t; M_t^{(K)})$ and $V_{\boldsymbol{\pi}_t}(\bar{\boldsymbol{S}}_1^t, \bar{\boldsymbol{X}}_1^t; M_t) = V_{\boldsymbol{\pi}_t}(S_1^t, X_1^t; M_t^{(1)})$. We
37 thus have

$$V_{\boldsymbol{\pi}_t}(\bar{\boldsymbol{S}}_1^t, \bar{\boldsymbol{X}}_1^t; M_t) - E_{\bar{\boldsymbol{X}}_K^t}^{M_t}[Y|\bar{\boldsymbol{S}}_K^t] = V_{\boldsymbol{\pi}_t}(\bar{\boldsymbol{S}}_1^t, \bar{\boldsymbol{X}}_1^t; M_t^{(1)}) - V_{\boldsymbol{\pi}_t}(\bar{\boldsymbol{S}}_K^t, \bar{\boldsymbol{X}}_K^t; M_t^{(K)})$$
$$= \sum_{k=1}^{K-1} V_{\boldsymbol{\pi}_t}(\bar{\boldsymbol{S}}_k^t, \bar{\boldsymbol{X}}_k^t; M_t^{(1)}) - V_{\boldsymbol{\pi}_t}(\bar{\boldsymbol{S}}_{k+1}^t, \bar{\boldsymbol{X}}_{k+1}^t; M_t^{(K)})$$
$$= \sum_{k=1}^{K-1} V_{\boldsymbol{\pi}_t}(\bar{\boldsymbol{S}}_k^t, \bar{\boldsymbol{X}}_k^t; M_t^{(1)}) - V_{\boldsymbol{\pi}_t}(\bar{\boldsymbol{S}}_{k+1}^t, \bar{\boldsymbol{X}}_k^t; M_t^{(K)}).$$

The last step is ensured by Eq. (21). We further have:

$$V_{\boldsymbol{\pi}_t}(\bar{\boldsymbol{S}}_1^t, \bar{\boldsymbol{X}}_1^t; M_t) - E_{\bar{\boldsymbol{X}}_K^t}^{M_t}[Y|\bar{\boldsymbol{S}}_K^t] = \sum_{k=1}^{K-1} V_{\boldsymbol{\pi}_t}(\bar{\boldsymbol{S}}_k^t, \bar{\boldsymbol{X}}_k^t; M_t^{(k)}) - V_{\boldsymbol{\pi}_t}(\bar{\boldsymbol{S}}_k^t, \bar{\boldsymbol{X}}_k^t; M_t^{(k+1)})$$
$$+ \sum_{k=1}^{K-1} V_{\boldsymbol{\pi}_t}(\bar{\boldsymbol{S}}_k^t, \bar{\boldsymbol{X}}_k^t; M_t^{(k+1)}) - V_{\boldsymbol{\pi}_t}(\bar{\boldsymbol{S}}_{k+1}^t, \bar{\boldsymbol{X}}_k^t; M_t^{(k+1)}).$$

Eq. (18) can thus be written as:

$$\sum_{t \in L_\epsilon} \left( V_{\boldsymbol{\pi}_t}(M_t) - E_{\bar{\boldsymbol{X}}_K^t}^{M_t}[Y|\bar{\boldsymbol{S}}_K^t] \right) = \sum_{k=1}^{K-1} \sum_{t \in L_\epsilon} V_{\boldsymbol{\pi}_t}(\bar{\boldsymbol{S}}_k^t, \bar{\boldsymbol{X}}_k^t; M_t^{(k)}) - V_{\boldsymbol{\pi}_t}(\bar{\boldsymbol{S}}_k^t, \bar{\boldsymbol{X}}_k^t; M_t^{(k+1)}) + \sum_{t \in L_\epsilon} Z_t,$$

where $Z_t$ is defined as

$$Z_t = V_{\boldsymbol{\pi}^*}(M^*) - V_{\boldsymbol{\pi}^*}(\bar{\boldsymbol{S}}_1^t; M) + \sum_{k=1}^{K-1} V_{\boldsymbol{\pi}_t}(\bar{\boldsymbol{S}}_k^t, \bar{\boldsymbol{X}}_k^t; M_t^{(k+1)}) - V_{\boldsymbol{\pi}_t}(\bar{\boldsymbol{S}}_{k+1}^t, \bar{\boldsymbol{X}}_k^t; M_t^{(k+1)})$$

By Eq. (22) and basic probabilistic operations,

$$V_{\boldsymbol{\pi}_t}(\bar{\boldsymbol{S}}_k^t, \bar{\boldsymbol{X}}_k^t; M_t^{(k)}) - V_{\boldsymbol{\pi}_t}(\bar{\boldsymbol{S}}_k^t, \bar{\boldsymbol{X}}_k^t; M_t^{(k+1)})$$
$$= \sum_{s_{k+1}} (P^{M_t}(s_{k+1}|\bar{\boldsymbol{S}}_k, \bar{\boldsymbol{X}}_k) - P(s_{k+1}|\bar{\boldsymbol{S}}_k, \bar{\boldsymbol{X}}_k)) V_{\boldsymbol{\pi}_t}(s_{k+1}, \bar{\boldsymbol{S}}_k^t, \bar{\boldsymbol{X}}_k^t; M_t)$$
$$\leq \left\| P_{\bar{\boldsymbol{x}}_k}^{M_t}(\cdot|\bar{\boldsymbol{s}}_k) - P_{\bar{\boldsymbol{x}}_k}(\cdot|\bar{\boldsymbol{s}}_k) \right\|_1 \max_{s_{k+1}} V_{\boldsymbol{\pi}_t}(s_{k+1}, \bar{\boldsymbol{S}}_k^t, \bar{\boldsymbol{X}}_k^t; M_t)$$
$$\leq 2\sqrt{6|\mathcal{S}_{k+1}| \log(2K|\bar{\boldsymbol{S}}_k||\bar{\boldsymbol{\mathcal{X}}}_k|T/\delta)} \frac{1}{\sqrt{\max\{1, N^t(\bar{\boldsymbol{S}}_k^t, \bar{\boldsymbol{X}}_k^t)}}$$

The last step follows from Eq. (16). From results in [4, D], we have

$$\sum_{t \in L_\epsilon} \frac{1}{\sqrt{\max\{1, N^t(\bar{\boldsymbol{S}}_k^t, \bar{\boldsymbol{X}}_k^t)\}}} \leq (\sqrt{2}+1)\sqrt{T_\epsilon |\bar{\boldsymbol{S}}_k||\bar{\boldsymbol{\mathcal{X}}}_k|}.$$

This implies:

$$\sum_{t \in L_\epsilon} \sum_{k=1}^{K-1} V_{\boldsymbol{\pi}_t}(\bar{\boldsymbol{S}}_k^t, \bar{\boldsymbol{X}}_k^t; M_t^{(k)}) - V_{\boldsymbol{\pi}_t}(\bar{\boldsymbol{S}}_k^t, \bar{\boldsymbol{X}}_k^t; M_t^{(k+1)})$$
$$\leq \sum_{k=1}^{K-1} 2(\sqrt{2}+1)\sqrt{6T_\epsilon |\bar{\boldsymbol{S}}_{k+1}||\bar{\boldsymbol{\mathcal{X}}}_k| \log(2K|\bar{\boldsymbol{S}}_k||\bar{\boldsymbol{\mathcal{X}}}_k|T/\delta)}$$
$$\leq 2(\sqrt{2}+1)(K-1)\sqrt{6T_\epsilon |\boldsymbol{\mathcal{S}}||\boldsymbol{\mathcal{X}}| \log(2K|\boldsymbol{\mathcal{S}}||\boldsymbol{\mathcal{X}}|T/\delta)} \tag{23}$$

Let $\mathcal{H}^t$ denote the history up to episode $t$, i.e., $\{\bar{\boldsymbol{X}}_K^1, \bar{\boldsymbol{S}}_K^1, Y^1, \ldots, \bar{\boldsymbol{X}}_K^t, \bar{\boldsymbol{S}}_K^t, Y^t\}$. Since $|Z_t| \leq K$ and $E[Z_{t+1}|\mathcal{H}_t] = 0$, $\{Z_t : t \in L_\epsilon\}$ is a sequence of martingale differences. By Azuma-Hoeffding inequality [3], we have, with probability at least $1 - \frac{\delta}{8T^2}$,

$$\sum_{t \in L_\epsilon} Z_t \leq K\sqrt{6T_\epsilon \log(2T/\delta)} \tag{24}$$

Since $\sum_{T=1}^{\infty} \frac{1}{8T^2} \leq \frac{\pi^2}{48}\delta < \frac{\delta}{4}$, the above inequality holds with probability $1 - \frac{\delta}{4}$ for all $T > 1$. Eqs. (23) and (24) combined give

$$\sum_{t \in L_\epsilon} \left( V_{\boldsymbol{\pi}^*}(M^*) - E_{\bar{\boldsymbol{X}}_K^t}^{M_t}[Y|\bar{\boldsymbol{S}}_K^t] \right)$$
$$\leq 2(\sqrt{2}+1)(K-1)\sqrt{6T_\epsilon |\boldsymbol{\mathcal{S}}||\boldsymbol{\mathcal{X}}| \log(2K|\boldsymbol{\mathcal{S}}||\boldsymbol{\mathcal{X}}|T/\delta)} + K\sqrt{6T_\epsilon \log(2T/\delta)} \tag{25}$$

49 **Bounding Eq. (19)** Since both $M^*, M_t$ are in the set $\mathcal{M}_t$,

$$E_{\bar{\boldsymbol{X}}_K^t}^{M_t}[Y|\bar{\boldsymbol{S}}_K^t] - E_{\bar{\boldsymbol{X}}_K^t}[Y|\bar{\boldsymbol{S}}_K^t] \leq \left| E_{\bar{\boldsymbol{x}}_K}^{M_t}[Y|\bar{\boldsymbol{s}}_K] - \hat{E}_{\bar{\boldsymbol{x}}_K}^t[Y|\bar{\boldsymbol{s}}_K] \right| + \left| E_{\bar{\boldsymbol{X}}_K^t}[Y|\bar{\boldsymbol{S}}_K^t] - \hat{E}_{\bar{\boldsymbol{x}}_K}^t[Y|\bar{\boldsymbol{s}}_K] \right|$$

$$\leq 2\sqrt{2\log(2K|\boldsymbol{\mathcal{S}}||\boldsymbol{\mathcal{X}}|T/\delta)} \frac{1}{\sqrt{\max\{1, N^t(\bar{\boldsymbol{S}}_K^t, \bar{\boldsymbol{X}}_K^t)\}}}$$

50 The last step follows from Eq. (17). From results in [4, D], we have

$$\sum_{t \in L_\epsilon} \frac{1}{\sqrt{\max\{1, N^t(\bar{\boldsymbol{S}}_K^t, \bar{\boldsymbol{X}}_K^t)\}}} \leq (\sqrt{2}+1)\sqrt{T_\epsilon|\boldsymbol{\mathcal{S}}||\boldsymbol{\mathcal{X}}|}.$$

51 This implies

$$\sum_{t \in L_\epsilon} \left( E_{\bar{\boldsymbol{X}}_K^t}^{M_t}[Y|\bar{\boldsymbol{S}}_K^t] - E_{\bar{\boldsymbol{X}}_K^t}[Y|\bar{\boldsymbol{S}}_K^t] \right) \leq 2(\sqrt{2}+1)\sqrt{2T_\epsilon|\boldsymbol{\mathcal{S}}||\boldsymbol{\mathcal{X}}|\log(2K|\boldsymbol{\mathcal{S}}||\boldsymbol{\mathcal{X}}|T/\delta)} \qquad (26)$$

52 **Bounding Eq. (20)** By Lem. 2, we have with probability at least $1 - \frac{\delta}{8T^2}$,

$$\sum_{t \in L_\epsilon} \left( E_{\bar{\boldsymbol{X}}_K^t}[Y|\bar{\boldsymbol{S}}_K^t] - Y^t \right) \leq \sqrt{\frac{3T_\epsilon \log(2T/\delta)}{2}} \qquad (27)$$

53 Since $\sum_{T=1}^{\infty} \frac{1}{8T^2} \leq \frac{\pi^2}{48}\delta < \frac{\delta}{4}$, the above equation holds with probability $1 - \frac{\delta}{4}$ for any $T$.

54 Eqs. (25) to (27) together give that, with probability at least $1 - \frac{\delta}{2} - \frac{\delta}{4} - \frac{\delta}{4} = 1 - \delta$,

$$R_\epsilon(T) \leq (K-1)2(\sqrt{2}+1)\sqrt{6T_\epsilon|\boldsymbol{\mathcal{S}}||\boldsymbol{\mathcal{X}}|\log(2K|\boldsymbol{\mathcal{S}}||\boldsymbol{\mathcal{X}}|T/\delta)} + K\sqrt{6T_\epsilon \log(2T/\delta)}$$

$$+ 2(\sqrt{2}+1)\sqrt{2T_\epsilon|\boldsymbol{\mathcal{S}}||\boldsymbol{\mathcal{X}}|\log(2K|\boldsymbol{\mathcal{S}}||\boldsymbol{\mathcal{X}}|T/\delta)} + \sqrt{\frac{3T_\epsilon \log(2T/\delta)}{2}}.$$

55 A quick simplification gives:

$$R_\epsilon(T) \leq 12K\sqrt{|\boldsymbol{\mathcal{S}}||\boldsymbol{\mathcal{X}}|T_\epsilon \log(2K|\boldsymbol{\mathcal{S}}||\boldsymbol{\mathcal{X}}|T/\delta)} + 4K\sqrt{T_\epsilon \log(2T/\delta)}. \qquad \square$$

56 **Theorem 1.** *Fix a $\delta \in (0,1)$. With probability (w.p.) of at least $1 - \delta$, it holds for any $T > 1$, the*
57 *regret of* UC-DTR *with parameter $\delta$ is bounded by*

$$R(T) \leq 12K\sqrt{|\boldsymbol{\mathcal{S}}||\boldsymbol{\mathcal{X}}|T \log(2K|\boldsymbol{\mathcal{S}}||\boldsymbol{\mathcal{X}}|T/\delta)} + 4K\sqrt{T \log(2T/\delta)}.$$

58 *Proof.* Fix $\epsilon = 0$. Naturally, $T_\epsilon = T$ and $R_\epsilon(T) = R(T)$. By Lem. 3,

$$R(T) \leq 12K\sqrt{|\boldsymbol{\mathcal{S}}||\boldsymbol{\mathcal{X}}|T \log(2K|\boldsymbol{\mathcal{S}}||\boldsymbol{\mathcal{X}}|T/\delta)} + 4K\sqrt{T \log(2T/\delta)}. \qquad \square$$

59 **Theorem 2.** *For any $T \geq 1$, with parameter $\delta = \frac{1}{T}$, the expected regret of* UC-DTR *is bounded by*

$$E[R(T)] \leq \max_{\boldsymbol{\pi} \in \boldsymbol{\Pi}^-} \left\{ \frac{33^2 K^2 |\boldsymbol{\mathcal{S}}||\boldsymbol{\mathcal{X}}| \log(T)}{\Delta_{\boldsymbol{\pi}}} + \frac{32}{\Delta_{\boldsymbol{\pi}}^3} + \frac{4}{\Delta_{\boldsymbol{\pi}}} \right\} + 1.$$

60 *Proof.* By Lem. 3 and a quick simplification, we have

$$R_\epsilon(T) \leq 23K\sqrt{|\boldsymbol{\mathcal{S}}||\boldsymbol{\mathcal{X}}|T_\epsilon \log(T/\delta)}.$$

61 Since $R_\epsilon(T) \geq \epsilon T_\epsilon$, $\epsilon T_\epsilon \leq 23K\sqrt{|\boldsymbol{\mathcal{S}}||\boldsymbol{\mathcal{X}}|T_\epsilon \log(T/\delta)}$, which implies

$$T_\epsilon \leq \frac{23^2 K^2 |\boldsymbol{\mathcal{S}}||\boldsymbol{\mathcal{X}}| \log(T/\delta)}{\epsilon^2}. \qquad (28)$$

62 This implies that, with probability at least $1 - \delta$,

$$R_\epsilon(T) \leq 23K\sqrt{|\boldsymbol{\mathcal{S}}||\boldsymbol{\mathcal{X}}|T_\epsilon \log(T/\delta)} = \frac{23^2 K^2 |\boldsymbol{\mathcal{S}}||\boldsymbol{\mathcal{X}}| \log(T/\delta)}{\epsilon}$$

Let $\Delta = \arg\min_{\boldsymbol{\pi} \in \boldsymbol{\Pi}^-} \Delta_{\boldsymbol{\pi}}$. Fix $\epsilon = \frac{\Delta}{2}$, $\delta = \frac{1}{T}$, we have

$$E[R_{\frac{\Delta}{2}}(T)] \leq \frac{33^2 K^2 |\boldsymbol{\mathcal{S}}||\boldsymbol{\mathcal{X}}|\log(T)}{\Delta} + 1. \tag{29}$$

We now only need to bound the regrets cumulated in the episodes that are not $\epsilon$-bad, which we call $\epsilon$-good. Let $\tilde{R}_\epsilon(T)$ denote the regret in episodes that are $\epsilon$-good. Let $\tilde{T}_\epsilon$ denote the total number of $\epsilon$-good episodes and let $\tilde{L}_\epsilon$ be indices of $\epsilon$-good episodes. Fix $\epsilon = \frac{\Delta}{2}$, for any $\epsilon$-good episode $t$, we have $V_{\boldsymbol{\pi}_t}(M^*) - Y^t < \epsilon$. Fix event $\tilde{T}_{\frac{\Delta}{2}} = t$,

$$\tilde{R}_\epsilon(T) = \sum_{i \in \tilde{L}_\epsilon} V_{\boldsymbol{\pi}^*}(M^*) - Y^i \leq t\frac{\Delta}{2}.$$

The above inequality is equivalent to

$$\sum_{i \in \tilde{L}_{\frac{\Delta}{2}}} V_{\boldsymbol{\pi}^*}(M^*) - V_{\boldsymbol{\pi}_i}(M^*) - Y^i \leq t\frac{\Delta}{2} - \sum_{i \in \tilde{L}_{\frac{\Delta}{2}}} V_{\boldsymbol{\pi}_i}(M^*)$$

$$\Rightarrow \sum_{i \in \tilde{L}_{\frac{\Delta}{2}}} \Delta_{\boldsymbol{\pi}_i} - Y^i \leq t\frac{\Delta}{2} - \sum_{i \in \tilde{L}_{\frac{\Delta}{2}}} V_{\boldsymbol{\pi}_i}(M^*)$$

$$\Rightarrow \sum_{i \in \tilde{L}_{\frac{\Delta}{2}}} \Delta - Y^i \leq t\frac{\Delta}{2} - \sum_{i \in \tilde{L}_{\frac{\Delta}{2}}} V_{\boldsymbol{\pi}_i}(M^*)$$

Since $|\tilde{L}_\epsilon| = \tilde{T}_{\frac{\Delta}{2}}$, we have

$$\tilde{T}_{\frac{\Delta}{2}} = t \Rightarrow \sum_{i \in \tilde{L}_{\frac{\Delta}{2}}} V_{\boldsymbol{\pi}_i}(M^*) - Y^i \leq -t\frac{\Delta}{2}. \tag{30}$$

We could thus bound $E[\tilde{R}_{\frac{\Delta}{2}}(T)]$ as

$$E[\tilde{R}_{\frac{\Delta}{2}}(T)] \leq \frac{\Delta}{2} E[\tilde{T}_{\frac{\Delta}{2}}(T)] \leq \frac{\Delta}{2} \sum_{t=1}^{T} t P(\tilde{T}_{\frac{\Delta}{2}} = t)$$

By Eq. (30), we further have

$$E[\tilde{R}_{\frac{\Delta}{2}}(T)] \leq \frac{\Delta}{2} \sum_{t=1}^{T} t P\left( \sum_{i \in \tilde{L}_{\frac{\Delta}{2}}} V_{\boldsymbol{\pi}_i}(M^*) - Y^i \leq -t\frac{\Delta}{2} \right)$$

Let $C_t = V_{\boldsymbol{\pi}_t}(M^*) - Y^t$. Since $|C_t| < 1$ and $E[C_{t+1}|\boldsymbol{\mathcal{H}}^t] = 0$, $\{C_i : i \in \tilde{L}_{\frac{\Delta}{2}}\}$ is a sequence of martingale differences. Applying Azuma-Hoeffding lemma gives,

$$P\left( \sum_{i \in \tilde{L}_{\frac{\Delta}{2}}} C_i \leq -t\frac{\Delta}{2} \right) \leq e^{-\frac{\Delta^2 t}{8}}.$$

Thus

$$E[\tilde{R}_{\frac{\Delta}{2}}(T)] \leq \frac{\Delta}{2} \sum_{t=1}^{T} t e^{-\frac{\Delta^2 t}{8}} \leq \frac{\Delta}{2} \frac{64}{\Delta^4} (\frac{\Delta^2}{8} + 1) e^{-\frac{\Delta^2}{8}}$$

which implies

$$E[\tilde{R}_{\frac{\Delta}{2}}(T)] \leq \frac{32}{\Delta^3} + \frac{4}{\Delta}. \tag{31}$$

Eqs. (29) and (31) together give:

$$E[R(T)] = E[R_{\frac{\Delta}{2}}(T)] + E[\tilde{R}_{\frac{\Delta}{2}}(T)] \leq \frac{33^2 K^2 |\boldsymbol{\mathcal{S}}||\boldsymbol{\mathcal{X}}|\log(T)}{\Delta} + \frac{32}{\Delta^3} + \frac{4}{\Delta} + 1$$

The right-hand side of the above inequality is a decreasing function regarding the gap $\Delta$. By a quick simplification, we prove the statement. $\qquad\square$

**Theorem 3.** *For any algorithm $\mathcal{A}$, any natural numbers $K \geq 1$, and $\left|\mathcal{S}^k\right| \geq 2, \left|\mathcal{X}^k\right| \geq 2$ for any $k \in \{1, \ldots, K\}$, there is a DTR $M$ with horizon $K$, state domains $\mathcal{S}$ and action domains $\mathcal{X}$, such that the expected regret of $\mathcal{A}$ after $T \geq |\mathcal{S}||\mathcal{X}|$ episodes is as least*

$$E[R(T)] \geq 0.05\sqrt{|\mathcal{S}||\mathcal{X}|T}.$$

*Proof.* The classic results in bandit literature [1, Thm. 5.1] shows that for each state sequence $_K$, there exists a bandit instance such that for any the total regret of any algorithm is lower bound by

$$E[R(T)] \geq 0.05 \sum_{\bar{s}_K} \sqrt{N(\bar{s}_K)|\mathcal{X}|},$$

where $N(\bar{s}_K)$ is the event count $\bar{S}_K = \bar{s}_K$ for all $T$ episodes. The lower bound in Thm. 3 is achieved when all states $_K$ are decided uniformly at random, i.e., $N(\bar{s}_K) = T/|\bar{\mathcal{S}}_K|$. $\qquad\square$

## Proofs of Theorems 4 to 6, Lemma 1, and Corollary 2

In this section, we provide proofs for the bounds on transition probabilities of DTRs. Our proofs build on the notion of counterfactual variables [6, Ch. 7.1] and axioms of "composition, effectiveness and reversibility" defined in [6, Ch. 7.3.1].

For a SCM $M$, arbitrary subsets of endogenous variables $\boldsymbol{X}, \boldsymbol{Y}$, the potential outcome of $\boldsymbol{Y}$ to intervention $do(\boldsymbol{x})$, denoted by $\boldsymbol{Y}_{\boldsymbol{x}}(\boldsymbol{u})$, is the solution for $Y$ with $\boldsymbol{U} = \boldsymbol{u}$ in the sub-model $M_{\boldsymbol{x}}$. It can be read as the counterfactual sentence "the value that $\boldsymbol{Y}$ would have obtained in situation $\boldsymbol{U} = \boldsymbol{u}$, had $\boldsymbol{X}$ been $\boldsymbol{x}$." Statistically, averaging $\boldsymbol{u}$ over the distribution $P(\boldsymbol{u})$ leads to the counterfactual variables $\boldsymbol{Y}_{\boldsymbol{x}}$. We denote $P(\boldsymbol{Y}_{\boldsymbol{x}})$ a distribution over counterfactual variables $\boldsymbol{Y}_{\boldsymbol{x}}$. We use $P(\boldsymbol{y}_{\boldsymbol{x}})$ as a shorthand for probabilities $P(\boldsymbol{Y}_{\boldsymbol{x}} = \boldsymbol{y})$ when the identify of the counterfactual variables is clear.

We now introduce a family of DTRs which represent the exogenous variables $\boldsymbol{U}$ using partitions defined by the corresponding counterfactual variables. For any $k = 1, \ldots, K - 1$, let $S_{k+1_{\bar{\boldsymbol{x}}_k}}$ denote a set of counterfactual variables $\{S_{k+1_{\bar{\boldsymbol{x}}_k}} : \bar{\boldsymbol{x}}_k \in \bar{\mathcal{X}}_k\}$. Similarly, let $Y_{\bar{\mathcal{X}}_K}$ denote a set $\{Y_{\bar{\boldsymbol{x}}_K} : \bar{\boldsymbol{x}}_K \in \bar{\mathcal{X}}_K\}$. Further, we define $\bar{\boldsymbol{S}}_{k+1_{\bar{\mathcal{X}}_k}}$ a set $\{S_1, S_{2_{\bar{\mathcal{X}}_1}}, \ldots, S_{k+1_{\bar{\mathcal{X}}_k}}\}$.

**Definition 1** (Counterfactual DTR). A counterfactual dynamic treatment regime is a DTR $\langle \boldsymbol{U}, \{\bar{\boldsymbol{X}}_K, \bar{\boldsymbol{S}}_K, Y\}, \boldsymbol{F}, P(\boldsymbol{u})\rangle$ where for $k = 2, \ldots, K$,

- The exogenous variables $\boldsymbol{U} = \{\bar{\boldsymbol{X}}_K, \bar{\boldsymbol{S}}_{K_{\bar{\mathcal{X}}_{K-1}}}, Y_{\bar{\mathcal{X}}_K}\}$;

- Values of $S_1, \bar{\boldsymbol{X}}_K$ are drawn from $P(\bar{\boldsymbol{X}}_K, \bar{\boldsymbol{S}}_{K_{\bar{\mathcal{X}}_{K-1}}}, Y_{\bar{\mathcal{X}}_K})$;

- Values of $S_k$ are decided by a function $S_k \leftarrow \tau_k(S_{k_{\bar{\mathcal{X}}_{k-1}}}, \bar{\boldsymbol{X}}_{k-1}) = S_{k_{\bar{\mathcal{X}}_{k-1}}}$;

- Values of $Y$ are decided by a function $Y \leftarrow r(Y_{\bar{\mathcal{X}}_K}, \bar{\boldsymbol{X}}_K) = Y_{\bar{\boldsymbol{X}}_K}$.

Give observational distribution $P(\bar{s}_K, \bar{\boldsymbol{x}}_K, y) > 0$, we next construct a family of counterfactual DTRs $\mathcal{M}_{\text{OBS}}$ that are compatible with the observational distribution, i.e., for any $M \in \mathcal{M}_{\text{OBS}}$, $P^M(\bar{s}_K, \bar{\boldsymbol{x}}_K, y) = P(\bar{s}_K, \bar{\boldsymbol{x}}_K, y)$. First, any $M \in \mathcal{M}_{\text{OBS}}$, its exogenous distribution $P^M(\bar{\boldsymbol{X}}_K, \bar{\boldsymbol{S}}_{K_{\bar{\mathcal{X}}_{K-1}}}, Y_{\bar{\mathcal{X}}_K})$ must satisfy the following decomposition:

$$P^M(\bar{\boldsymbol{X}}_K, \bar{\boldsymbol{S}}_{K_{\bar{\mathcal{X}}_{K-1}}}, Y_{\bar{\mathcal{X}}_K}) = P^M(s_1) \prod_{\bar{\boldsymbol{x}}_K^y \in \bar{\mathcal{X}}_K} P^M(Y_{\bar{\boldsymbol{x}}_K^y}|\bar{\boldsymbol{S}}_{K_{\bar{\boldsymbol{x}}_{K-1}}}, \bar{\boldsymbol{X}}_K)P^M(\bar{\boldsymbol{X}}_K|\bar{\boldsymbol{S}}_{K_{\bar{\boldsymbol{x}}_{K-1}}}, \bar{\boldsymbol{X}}_{K-1})$$

$$\cdot \prod_{k=1}^{K-1} \prod_{\bar{\boldsymbol{x}}_k^{k+1} \in \bar{\mathcal{X}}_k} P^M(S_{k+1_{\bar{\boldsymbol{x}}_k^{k+1}}}|\bar{\boldsymbol{S}}_{k_{\bar{\boldsymbol{x}}_{k-1}}}, \bar{\boldsymbol{x}}_k)P^M(\bar{\boldsymbol{X}}_k|\bar{\boldsymbol{S}}_{k_{\bar{\boldsymbol{x}}_{k-1}}}, \bar{\boldsymbol{X}}_{k-1}).$$

Among quantities in the above equation, we define factors $P^M(s_1)$ as the observational probabilities $P(s_1)$, i.e, $P^M(s_1) = P(s_1)$. We further define conditional probabilities

$$P^M(y_{\bar{\boldsymbol{x}}_K}|\bar{\boldsymbol{s}}_{K_{\bar{\boldsymbol{x}}_{K-1}}}, \bar{\boldsymbol{x}}_K) = P(y|\bar{s}_K, \bar{\boldsymbol{x}}_K), \qquad P^M(\bar{\boldsymbol{x}}_K|\bar{\boldsymbol{s}}_{K_{\bar{\boldsymbol{x}}_{K-1}}}, \bar{\boldsymbol{x}}_{K-1}) = P(\bar{\boldsymbol{x}}_K|\bar{s}_K, \bar{\boldsymbol{x}}_{K-1}),$$

$$P^M(s_{k+1_{\bar{\boldsymbol{x}}_k}}|\bar{\boldsymbol{s}}_{k_{\bar{\boldsymbol{x}}_{k-1}}}, \bar{\boldsymbol{x}}_k) = P(s_{k+1}|\bar{s}_k, \bar{\boldsymbol{x}}_k), \quad P^M(\bar{\boldsymbol{x}}_k|\bar{\boldsymbol{s}}_{k_{\bar{\boldsymbol{x}}_{k-1}}}, \bar{\boldsymbol{x}}_{k-1}) = P(\bar{\boldsymbol{x}}_k|\bar{s}_k, \bar{\boldsymbol{x}}_{k-1}).$$

112 Other factors can be arbitrary conditional probabilities. It is verifiable that for any $M \in \mathcal{M}_{\text{OBS}}$,
113 $P^M(\bar{\boldsymbol{s}}_K, \bar{\boldsymbol{x}}_K, y) = P(\bar{\boldsymbol{s}}_K, \bar{\boldsymbol{x}}_K, y)$. To witness,

$$
P^M(\bar{\boldsymbol{S}}_K, \bar{\boldsymbol{X}}_K, Y) = \sum_{k=1}^{K-1} \sum_{\{Y_{\bar{\boldsymbol{x}}_K^y} : \bar{\boldsymbol{x}}_K^y \neq \bar{\boldsymbol{x}}_K\}} \sum_{\{S_{k+1_{\bar{\boldsymbol{x}}_k^{k+1}}} : \bar{\boldsymbol{x}}_k^{k+1} \neq \bar{\boldsymbol{x}}_k\}} P^M(\bar{\boldsymbol{X}}_K, \bar{\boldsymbol{S}}_{K_{\bar{\boldsymbol{x}}_{K-1}}}, Y_{\bar{\boldsymbol{x}}_K})
$$

$$
= P^M(s_1) \prod_{\bar{\boldsymbol{x}}_K^y \in \bar{\boldsymbol{\mathcal{X}}}_K} \sum_{\{Y_{\bar{\boldsymbol{x}}_K^y} : \bar{\boldsymbol{x}}_K^y \neq \bar{\boldsymbol{x}}_K\}} P^M(Y_{\bar{\boldsymbol{x}}_K^y} | \bar{\boldsymbol{S}}_{K_{\bar{\boldsymbol{x}}_{K-1}}}, \bar{\boldsymbol{X}}_K) P^M(\bar{\boldsymbol{X}}_K | \bar{\boldsymbol{S}}_{K_{\bar{\boldsymbol{x}}_{K-1}}}, \bar{\boldsymbol{x}}_{K-1})
$$

$$
\cdot \prod_{k=1}^{K-1} \prod_{\bar{\boldsymbol{x}}_k^{k+1} \in \bar{\boldsymbol{\mathcal{X}}}_k} \sum_{\{S_{k+1_{\bar{\boldsymbol{x}}_k^{k+1}}} : \bar{\boldsymbol{x}}_k^{k+1} \neq \bar{\boldsymbol{x}}_k\}} P^M(S_{k+1_{\bar{\boldsymbol{x}}_k^{k+1}}} | \bar{\boldsymbol{S}}_{k_{\bar{\boldsymbol{x}}_{k-1}}}, \bar{\boldsymbol{X}}_k) P^M(\bar{\boldsymbol{X}}_k | \bar{\boldsymbol{S}}_{k_{\bar{\boldsymbol{x}}_{k-1}}}, \bar{\boldsymbol{X}}_{k-1})
$$

$$
= P^M(S_1) P^M(Y_{\bar{\boldsymbol{x}}_K} | \bar{\boldsymbol{S}}_{K_{\bar{\boldsymbol{x}}_{K-1}}}, \bar{\boldsymbol{X}}_K) P^M(\bar{\boldsymbol{X}}_K | \bar{\boldsymbol{S}}_{K_{\bar{\boldsymbol{x}}_{K-1}}}, \bar{\boldsymbol{X}}_{K-1})
$$

$$
\cdot \prod_{k=1}^{K-1} P^M(S_{k+1_{\bar{\boldsymbol{x}}_k}} | \bar{\boldsymbol{S}}_{k_{\bar{\boldsymbol{x}}_{k-1}}}, \bar{\boldsymbol{x}}_k) P^M(\bar{\boldsymbol{X}}_k | \bar{\boldsymbol{S}}_{k_{\bar{\boldsymbol{x}}_{k-1}}}, \bar{\boldsymbol{X}}_{k-1}).
$$

114 By definitions of $\mathcal{M}_{\text{OBS}}$, we thus have that, for any $\bar{\boldsymbol{s}}_K, \bar{\boldsymbol{x}}_K, y$,

$$
P^M(\bar{\boldsymbol{s}}_K, \bar{\boldsymbol{x}}_K, y) = P(s_1) P(y | \bar{\boldsymbol{s}}_K, \bar{\boldsymbol{x}}_K) P(\bar{\boldsymbol{x}}_K | \bar{\boldsymbol{s}}_K, \bar{\boldsymbol{x}}_{K-1}) \prod_{k=1}^{K-1} P(s_{k+1} | \bar{\boldsymbol{s}}_k, \bar{\boldsymbol{x}}_k) P(\bar{\boldsymbol{x}}_k | \bar{\boldsymbol{s}}_k, \bar{\boldsymbol{x}}_{k-1})
$$

$$
= P(\bar{\boldsymbol{s}}_K, \bar{\boldsymbol{x}}_K, y).
$$

115 We will now use the constructions of $\mathcal{M}_{\text{OBS}}$ to prove the non-identifiability of $P_{\bar{\boldsymbol{x}}_K}(\bar{\boldsymbol{s}}_K, y)$ in DTRs.

116 **Theorem 4.** *Given $P(\bar{\boldsymbol{s}}_K, \bar{\boldsymbol{x}}_K, y) > 0$, there exists DTRs $M_1, M_2$ such that $P^{M_1}(\bar{\boldsymbol{s}}_K, \bar{\boldsymbol{x}}_K, y) =$*
117 *$P^{M_2}(\bar{\boldsymbol{s}}_K, \bar{\boldsymbol{x}}_K, y) = P(\bar{\boldsymbol{s}}_K, \bar{\boldsymbol{x}}_K, y)$ while $P_{\bar{\boldsymbol{x}}_K}^{M_1}(\bar{\boldsymbol{s}}_K, y) \neq P_{\bar{\boldsymbol{x}}_K}^{M_2}(\bar{\boldsymbol{s}}_K, y)$.*

118 *Proof.* We define two counterfactual DTRs $M_1, M_2 \in \mathcal{M}_{\text{OBS}}$ that are compatible with the observa-
119 tional distribution $P(\bar{\boldsymbol{s}}_K, \bar{\boldsymbol{x}}_K, y)$. If $K = 1$, for any $y, s_1, x_1$ and any $x_1^y \neq x_1$, we define

$$
P^{M_1}(y_{x_1^y} | s_1, x_1) = 0, \qquad\qquad P^{M_2}(y_{x_1^y} | s_1, x_1) = 1
$$

120 It is verifiable that

$$
P_{x_1}^{M_1}(s_1, y) = P(s_1, x_1, y), \qquad P_{x_1}^{M_2}(s_1, y) = P(s_1, x_1, y) + (1 - P(x_1 | s_1)) P(s_1)
$$

121 Since $P(\bar{\boldsymbol{s}}_K, \bar{\boldsymbol{x}}_K, y) > 0$, we have $P_{x_1}^{M_2}(s_1, y) \neq P_{x_1}^{M_1}(s_1, y)$.

122 We now consider the case where $K > 1$. For any $\bar{\boldsymbol{x}}_K, \bar{\boldsymbol{s}}_K, y$, and any $\bar{\boldsymbol{x}}_K^y \neq \bar{\boldsymbol{x}}_K$, we define

$$
P^{M_1}(y_{\bar{\boldsymbol{x}}_K^y} | \bar{\boldsymbol{s}}_{K_{\bar{\boldsymbol{x}}_{K-1}}}, \bar{\boldsymbol{x}}_K) = 0 \tag{32}
$$

123 By definitions, $P_{\bar{\boldsymbol{x}}_K}^{M_1}(\bar{\boldsymbol{s}}_K, y)$ is equal to the counterfactual quantities $P^{M_1}(\bar{\boldsymbol{s}}_{K_{\bar{\boldsymbol{x}}_{K-1}}}, y_{\bar{\boldsymbol{x}}_K})$. Thus,

$$
P_{\bar{\boldsymbol{x}}_K}^{M_1}(\bar{\boldsymbol{s}}_K, y) = P^{M_1}(\bar{\boldsymbol{s}}_{K_{\bar{\boldsymbol{x}}_{K-1}}}, y_{\bar{\boldsymbol{x}}_K}, \bar{\boldsymbol{x}}_K) + \sum_{\bar{\boldsymbol{x}}_K' \neq \bar{\boldsymbol{x}}_K} P^{M_1}(\bar{\boldsymbol{s}}_{K_{\bar{\boldsymbol{x}}_{K-1}}}, y_{\bar{\boldsymbol{x}}_K}, \bar{\boldsymbol{x}}_K')
$$

$$
= P^{M_1}(\bar{\boldsymbol{s}}_{K_{\bar{\boldsymbol{x}}_{K-1}}}, y_{\bar{\boldsymbol{x}}_K}, \bar{\boldsymbol{x}}_K) + \sum_{\bar{\boldsymbol{x}}_K' \neq \bar{\boldsymbol{x}}_K} P^{M_1}(y_{\bar{\boldsymbol{x}}_K} | \bar{\boldsymbol{s}}_{K_{\bar{\boldsymbol{x}}_{K-1}}}, \bar{\boldsymbol{x}}_K') P^{M_1}(\bar{\boldsymbol{s}}_{K_{\bar{\boldsymbol{x}}_{K-1}}}, \bar{\boldsymbol{x}}_K')
$$

124 By the composition axiom, $\bar{\boldsymbol{S}}_{K_{\bar{\boldsymbol{x}}_{K-1}}} = \bar{\boldsymbol{S}}_K, Y_{\bar{\boldsymbol{x}}_K} = Y$ if $\bar{\boldsymbol{X}}_K = \bar{\boldsymbol{x}}_K$. Thus,
125 $P^{M_1}(\bar{\boldsymbol{s}}_{K_{\bar{\boldsymbol{x}}_{K-1}}}, y_{\bar{\boldsymbol{x}}_K}, \bar{\boldsymbol{x}}_K) = P^{M_1}(\bar{\boldsymbol{s}}_K, y, \bar{\boldsymbol{x}}_K)$. Since $M_1 \in \mathcal{M}_{\text{OBS}}$, $P^{M_1}(\bar{\boldsymbol{s}}_K, y, \bar{\boldsymbol{x}}_K) =$
126 $P(\bar{\boldsymbol{s}}_K, y, \bar{\boldsymbol{x}}_K)$. Together with Eq. (32), we can obtain

$$
P_{\bar{\boldsymbol{x}}_K}^{M_1}(\bar{\boldsymbol{s}}_K, y) = P(\bar{\boldsymbol{s}}_K, \bar{\boldsymbol{x}}_K, y).
$$

127 As for $M_2$, for any $\bar{\boldsymbol{x}}_{K-1}^K \neq \bar{\boldsymbol{x}}_{k-1}$, we define its factor

$$
P^{M_2}(s_{K_{\bar{\boldsymbol{x}}_{K-1}^K}} | \bar{\boldsymbol{s}}_{K-1_{\bar{\boldsymbol{x}}_{K-2}}}, \bar{\boldsymbol{x}}_{K-1}) = 0
$$

128  The above equation implies that for any $\bar{\boldsymbol{x}}'_{K-1} \neq \bar{\boldsymbol{x}}_{K-1}$,

$$P^{M_2}(\bar{\boldsymbol{s}}_{K_{\bar{\boldsymbol{x}}_{K-1}}}, y_{\bar{\boldsymbol{x}}_K}, \bar{\boldsymbol{x}}'_{K-1})$$
$$= P^{M_2}(y_{\bar{\boldsymbol{x}}_K}|\bar{\boldsymbol{s}}_{K_{\bar{\boldsymbol{x}}_{K-1}}}, \bar{\boldsymbol{x}}'_{K-1}) P^{M_2}(s_{K_{\bar{\boldsymbol{x}}_{K-1}}}|\bar{\boldsymbol{s}}_{K-1_{\bar{\boldsymbol{x}}_{K-2}}}, \bar{\boldsymbol{x}}'_{K-1}) P^{M_2}(\bar{\boldsymbol{s}}_{K-1_{\bar{\boldsymbol{x}}_{K-2}}}, \bar{\boldsymbol{x}}'_{K-1})$$
$$= 0 \tag{33}$$

129  For any $\bar{\boldsymbol{x}}^y_K \neq \bar{\boldsymbol{x}}_K$, we define

$$P^{M_2}(y_{\bar{\boldsymbol{x}}^y_K}|\bar{\boldsymbol{s}}_{K_{\bar{\boldsymbol{x}}_{K-1}}}, \bar{\boldsymbol{x}}_K) = 1 \tag{34}$$

130  We will now show that the above equation implies that for any $x'_K \neq x_K$,

$$P^{M_2}(\bar{\boldsymbol{s}}_{K_{\bar{\boldsymbol{x}}_{K-1}}}, y_{\bar{\boldsymbol{x}}_k}, x'_K, \bar{\boldsymbol{x}}_{K-1}) = P(\bar{\boldsymbol{s}}_K, \bar{\boldsymbol{x}}_{K-1}). \tag{35}$$

131  We first write $P^{M_2}(\bar{\boldsymbol{s}}_{K_{\bar{\boldsymbol{x}}_{K-1}}}, y_{\bar{\boldsymbol{x}}_k}, x'_K, \bar{\boldsymbol{x}}_{K-1})$ as:

$$P^{M_2}(\bar{\boldsymbol{s}}_{K_{\bar{\boldsymbol{x}}_{K-1}}}, y_{\bar{\boldsymbol{x}}_k}, x'_K, \bar{\boldsymbol{x}}_{K-1}) = P^{M_2}(y_{\bar{\boldsymbol{x}}_k}|\bar{\boldsymbol{s}}_{K_{\bar{\boldsymbol{x}}_{K-1}}}, x'_k, \bar{\boldsymbol{x}}_{K-1}) P^{M_2}(\bar{\boldsymbol{s}}_{K_{\bar{\boldsymbol{x}}_{K-1}}}, x'_k, \bar{\boldsymbol{x}}_{K-1})$$

132  It is immediate from Eq. (34) that

$$P^{M_2}(\bar{\boldsymbol{s}}_{K_{\bar{\boldsymbol{x}}_{K-1}}}, y_{\bar{\boldsymbol{x}}_k}, X_k \neq x_k, \bar{\boldsymbol{x}}_{K-1}) = P^{M_2}(\bar{\boldsymbol{s}}_{K_{\bar{\boldsymbol{x}}_{K-1}}}, \bar{\boldsymbol{x}}_{K-1}).$$

133  By the composition axiom, $\bar{\boldsymbol{S}}_{K_{\bar{\boldsymbol{x}}_{K-1}}} = \bar{\boldsymbol{S}}_K$ if $\bar{\boldsymbol{X}}_{K-1} = \bar{\boldsymbol{x}}_{k-1}$. Since $M_2 \in \mathcal{M}_{\text{OBS}}$, we thus have:

$$P^{M_2}(\bar{\boldsymbol{s}}_{K_{\bar{\boldsymbol{x}}_{K-1}}}, y_{\bar{\boldsymbol{x}}_k}, X_k \neq x_k, \bar{\boldsymbol{x}}_{K-1}) = P^{M_2}(\bar{\boldsymbol{s}}_K, \bar{\boldsymbol{x}}_{K-1}) = P(\bar{\boldsymbol{s}}_K, \bar{\boldsymbol{x}}_{K-1}).$$

134  We now turn our attention to the interventional distribution $P^{M_2}_{\bar{\boldsymbol{x}}_K}(\bar{\boldsymbol{s}}_K, y)$. By expanding on $\bar{\boldsymbol{X}}_K$,

$$P^{M_2}_{\bar{\boldsymbol{x}}_K}(\bar{\boldsymbol{s}}_K, y) = P^{M_2}(\bar{\boldsymbol{s}}_{K_{\bar{\boldsymbol{x}}_{K-1}}}, y_{\bar{\boldsymbol{x}}_k}, \bar{\boldsymbol{x}}_K) + P^{M_2}(\bar{\boldsymbol{s}}_{K_{\bar{\boldsymbol{x}}_{K-1}}}, y_{\bar{\boldsymbol{x}}_k}, X_k \neq x_k, \bar{\boldsymbol{x}}_{K-1})$$
$$+ P^{M_2}(\bar{\boldsymbol{s}}_{K_{\bar{\boldsymbol{x}}_{K-1}}}, y_{\bar{\boldsymbol{x}}_k}, \bar{\boldsymbol{X}}_{K-1} \neq \bar{\boldsymbol{x}}_{K-1})$$

135  The above equation, together with Eqs. (33) and (35), gives:

$$P^{M_2}_{\bar{\boldsymbol{x}}_K}(\bar{\boldsymbol{s}}_K, y) = P^{M_2}(\bar{\boldsymbol{s}}_{K_{\bar{\boldsymbol{x}}_{K-1}}}, y_{\bar{\boldsymbol{x}}_k}, \bar{\boldsymbol{x}}_K) + P(\bar{\boldsymbol{s}}_K, \bar{\boldsymbol{x}}_{K-1}).$$

136  Again, by the composition axiom and $M_2 \in \mathcal{M}_{\text{OBS}}$,

$$P^{M_2}_{\bar{\boldsymbol{x}}_K}(\bar{\boldsymbol{s}}_K, y) = P^{M_2}(\bar{\boldsymbol{s}}_K, y, \bar{\boldsymbol{x}}_K) + P(\bar{\boldsymbol{s}}_K, \bar{\boldsymbol{x}}_{K-1}) = P(\bar{\boldsymbol{s}}_K, y, \bar{\boldsymbol{x}}_K) + P(\bar{\boldsymbol{s}}_K, \bar{\boldsymbol{x}}_{K-1}).$$

137  Since $P(\bar{\boldsymbol{s}}_K, \bar{\boldsymbol{x}}_{K-1}) > 0$, we have $P^{M_1}_{\bar{\boldsymbol{x}}_K}(\bar{\boldsymbol{s}}_K, y) \neq P^{M_2}_{\bar{\boldsymbol{x}}_K}(\bar{\boldsymbol{s}}_K, y)$, which proves the statement. $\square$

138  **Lemma 1.** *For a DTR, given $P(\bar{\boldsymbol{s}}_K, \bar{\boldsymbol{x}}_K, y)$, for any $k = 1, \ldots, K-1$,*

$$P_{\bar{\boldsymbol{x}}_k}(\bar{\boldsymbol{s}}_{k+1}) - P_{\bar{\boldsymbol{x}}_k}(\bar{\boldsymbol{s}}_k) \leq P(\bar{\boldsymbol{s}}_{k+1}, \bar{\boldsymbol{x}}_k) - P(\bar{\boldsymbol{s}}_k, \bar{\boldsymbol{x}}_k).$$

139  *Proof.* Note that $P_{\bar{\boldsymbol{x}}_k}(\bar{\boldsymbol{s}}_{k+1})$ can be written as the counterfactual quantity $P(\bar{\boldsymbol{s}}_{k+1_{\bar{\boldsymbol{x}}_k}})$. For any set of
140  variables $\boldsymbol{V}$, let $\neg\boldsymbol{v}$ denote an event $\boldsymbol{V} \neq \boldsymbol{v}$. $P_{\bar{\boldsymbol{x}}_k}(\bar{\boldsymbol{s}}_{k+1})$ could thus be written as:

$$P_{\bar{\boldsymbol{x}}_k}(\bar{\boldsymbol{s}}_{k+1}) = P(\bar{\boldsymbol{s}}_{k+1_{\bar{\boldsymbol{x}}_k}}, \bar{\boldsymbol{x}}_k) + P(\bar{\boldsymbol{s}}_{k+1_{\bar{\boldsymbol{x}}_k}}, \neg x_k, \bar{\boldsymbol{x}}_{k-1}) + P(\bar{\boldsymbol{s}}_{k+1_{\bar{\boldsymbol{x}}_k}}, \neg\bar{\boldsymbol{x}}_{k-1}),$$

141  By the composition axiom, $\bar{\boldsymbol{S}}_{k+1_{\bar{\boldsymbol{x}}_k}} = \bar{\boldsymbol{S}}_{k+1}$ if $\bar{\boldsymbol{X}}_k = \bar{\boldsymbol{x}}_k$. So,

$$P_{\bar{\boldsymbol{x}}_k}(\bar{\boldsymbol{s}}_{k+1}) = P(\bar{\boldsymbol{s}}_{k+1}, \bar{\boldsymbol{x}}_k) + P(\bar{\boldsymbol{s}}_{k+1_{\bar{\boldsymbol{x}}_k}}, \neg x_k, \bar{\boldsymbol{x}}_{k-1}) + P(\bar{\boldsymbol{s}}_{k+1_{\bar{\boldsymbol{x}}_k}}, \neg\bar{\boldsymbol{x}}_{k-1})$$
$$\leq P(\bar{\boldsymbol{s}}_{k+1}, \bar{\boldsymbol{x}}_k) + P(\bar{\boldsymbol{s}}_{k_{\bar{\boldsymbol{x}}_k}}, \neg x_k, \bar{\boldsymbol{x}}_{k-1}) + P(\bar{\boldsymbol{s}}_{k_{\bar{\boldsymbol{x}}_k}}, \neg\bar{\boldsymbol{x}}_{k-1})$$
$$= P(\bar{\boldsymbol{s}}_{k+1}, \bar{\boldsymbol{x}}_k) + P(\bar{\boldsymbol{s}}_{k_{\bar{\boldsymbol{x}}_k}}, \bar{\boldsymbol{x}}_{k-1}) - P(\bar{\boldsymbol{s}}_{k_{\bar{\boldsymbol{x}}_k}}, \bar{\boldsymbol{x}}_k) + P(\bar{\boldsymbol{s}}_{k_{\bar{\boldsymbol{x}}_k}}) - P(\bar{\boldsymbol{s}}_{k_{\bar{\boldsymbol{x}}_k}}, \bar{\boldsymbol{x}}_{k-1})$$
$$= P(\bar{\boldsymbol{s}}_{k_{\bar{\boldsymbol{x}}_k}}) + P(\bar{\boldsymbol{s}}_{k+1}, \bar{\boldsymbol{x}}_k) - P(\bar{\boldsymbol{s}}_{k_{\bar{\boldsymbol{x}}_k}}, \bar{\boldsymbol{x}}_k).$$

142  Again, by the composition axiom, $\bar{\boldsymbol{S}}_{k_{\bar{\boldsymbol{x}}_k}} = \bar{\boldsymbol{S}}_k$ if $\bar{\boldsymbol{X}}_k = \bar{\boldsymbol{x}}_k$. Since $P(\bar{\boldsymbol{s}}_{k_{\bar{\boldsymbol{x}}_k}}) = P_{\bar{\boldsymbol{x}}_k}(\bar{\boldsymbol{s}}_k)$,

$$P_{\bar{\boldsymbol{x}}_k}(\bar{\boldsymbol{s}}_{k+1}) \leq P_{\bar{\boldsymbol{x}}_k}(\bar{\boldsymbol{s}}_k) + P(\bar{\boldsymbol{s}}_{k+1}, \bar{\boldsymbol{x}}_k) - P(\bar{\boldsymbol{s}}_k, \bar{\boldsymbol{x}}_k)$$

143  Rearranging the above equation proves the statement. $\square$

144  **Lemma 4.** *For a DTR, given $P(\bar{\boldsymbol{s}}_K, \bar{\boldsymbol{x}}_K, y)$, for any $k = 0, \ldots, K-1$,*

$$P_{\bar{\boldsymbol{x}}_k}(\bar{\boldsymbol{s}}_{k+1}) \leq \Gamma(\bar{\boldsymbol{s}}_{k+1}, \bar{\boldsymbol{x}}_k),$$

145  *where $\Gamma(\bar{\boldsymbol{s}}_{k+1}, \bar{\boldsymbol{x}}_k) = P(\bar{\boldsymbol{s}}_{k+1}, \bar{\boldsymbol{x}}_k) - P(\bar{\boldsymbol{s}}_k, \bar{\boldsymbol{x}}_k) + \Gamma(\bar{\boldsymbol{s}}_k, \bar{\boldsymbol{x}}_{k-1})$ and $\Gamma(s_1) = P(s_1)$.*

146  *Proof.* We prove this statement by induction.

147   **Base Case:** $k = 0$   By definition, $\Gamma(s_1) = P(s_1)$. We thus have $P(s_1) \le \Gamma(s_1)$.

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

165   *Proof.* Without loss of generality, we assume that $K > 1$. We consider two counterfactual DTRs
166   $M_1, M_2 \in \mathcal{M}_{\text{OBS}}$ compatible with the observational distribution $P(\bar{\boldsymbol{s}}_K, \bar{\boldsymbol{x}}_K, y)$, which we define at
167   the beginning of this section. For all $i = 1, \ldots, k-1$, for any $\bar{\boldsymbol{x}}_i^{i+1} \ne \bar{\boldsymbol{x}}_i$, we define that for any
168   $M \in \{M_1, M_2\}$, its factors satisfy:

$$P^M(s_{i+1_{\bar{\boldsymbol{x}}_i^{i+1}}} | \bar{\boldsymbol{s}}_{i_{\bar{\boldsymbol{x}}_{i-1}}}, \bar{\boldsymbol{x}}_i) = 1. \tag{36}$$

169   Following a similar argument in Lem. 1, we will show that for any $M \in \{M_1, M_2\}$, for any
170   $i = 1, \ldots, k-1$,

$$P^M_{\bar{\boldsymbol{x}}_i}(\bar{\boldsymbol{s}}_{i+1}) - P^M_{\bar{\boldsymbol{x}}_i}(\bar{\boldsymbol{s}}_i) = P(\bar{\boldsymbol{s}}_{i+1}, \bar{\boldsymbol{x}}_i) - P(\bar{\boldsymbol{s}}_i, \bar{\boldsymbol{x}}_i). \tag{37}$$

171    By $P^M_{\bar{\boldsymbol{x}}_i}(\bar{\boldsymbol{s}}_{i+1}) = P^M(\bar{\boldsymbol{s}}_{i+1_{\bar{\boldsymbol{x}}_i}})$ and basic probabilistic operations,

$$P^M_{\bar{\boldsymbol{x}}_i}(\bar{\boldsymbol{s}}_{i+1}) = P^M(\bar{\boldsymbol{s}}_{i+1_{\bar{\boldsymbol{x}}_i}}, \bar{\boldsymbol{x}}_i) + P^M(\bar{\boldsymbol{s}}_{i+1_{\bar{\boldsymbol{x}}_i}}, X_i \neq x_i, \bar{\boldsymbol{x}}_{i-1}) + P^M(\bar{\boldsymbol{s}}_{i+1_{\bar{\boldsymbol{x}}_i}}, \bar{\boldsymbol{X}}_{i-1} \neq \bar{\boldsymbol{x}}_{i-1}).$$

172    By the composition axiom, $\bar{\boldsymbol{S}}_{i+1_{\bar{\boldsymbol{x}}_i}} = \bar{\boldsymbol{S}}_{i+1}$ if $\bar{\boldsymbol{X}}_i = \bar{\boldsymbol{x}}_i$. Since $M \in \mathcal{M}_{\text{OBS}}$, $P^M(\bar{\boldsymbol{s}}_{i+1}, \bar{\boldsymbol{x}}_i) =$
173    $P(\bar{\boldsymbol{s}}_{i+1}, \bar{\boldsymbol{x}}_i)$. Therefore,

$$P^M_{\bar{\boldsymbol{x}}_i}(\bar{\boldsymbol{s}}_{i+1}) = P(\bar{\boldsymbol{s}}_{i+1}, \bar{\boldsymbol{x}}_i) + P^M(\bar{\boldsymbol{s}}_{i+1_{\bar{\boldsymbol{x}}_i}}, X_i \neq x_i, \bar{\boldsymbol{x}}_{i-1}) + P^M(\bar{\boldsymbol{s}}_{i+1_{\bar{\boldsymbol{x}}_i}}, \bar{\boldsymbol{X}}_{i-1} \neq \bar{\boldsymbol{x}}_{i-1}),$$

$$= P(\bar{\boldsymbol{s}}_{i+1}, \bar{\boldsymbol{x}}_i) + \sum_{x'_i \neq x_i} P^M(s_{i+1_{\bar{\boldsymbol{x}}_i}} | \bar{\boldsymbol{s}}_{i_{\bar{\boldsymbol{x}}_{i-1}}}, x'_i, \bar{\boldsymbol{x}}_{i-1}) P(\bar{\boldsymbol{s}}_{i_{\bar{\boldsymbol{x}}_{i-1}}}, x'_i, \bar{\boldsymbol{x}}_{i-1})$$

$$+ \sum_{\bar{\boldsymbol{x}}'_{i-1} \neq \bar{\boldsymbol{x}}_{i-1}} P^M(s_{i+1_{\bar{\boldsymbol{x}}_i}} | \bar{\boldsymbol{s}}_{i_{\bar{\boldsymbol{x}}_{i-1}}}, x_i, \bar{\boldsymbol{x}}'_{i-1}) P(\bar{\boldsymbol{s}}_{i_{\bar{\boldsymbol{x}}_{i-1}}}, x_i, \bar{\boldsymbol{x}}'_{i-1})$$

174    By Eq. (36), $P^M(s_{i+1_{\bar{\boldsymbol{x}}_i}} | \bar{\boldsymbol{s}}_{i_{\bar{\boldsymbol{x}}_{i-1}}}, x'_i, \bar{\boldsymbol{x}}_{i-1}) = P^M(s_{i+1_{\bar{\boldsymbol{x}}_i}} | \bar{\boldsymbol{s}}_{i_{\bar{\boldsymbol{x}}_{i-1}}}, x_i, \bar{\boldsymbol{x}}'_{i-1}) = 1$, which gives

$$P^M_{\bar{\boldsymbol{x}}_i}(\bar{\boldsymbol{s}}_{i+1}) = P(\bar{\boldsymbol{s}}_{i+1}, \bar{\boldsymbol{x}}_i) + P^M(\bar{\boldsymbol{s}}_{i_{\bar{\boldsymbol{x}}_i}}, X_i \neq x_i, \bar{\boldsymbol{x}}_{i-1}) + P^M(\bar{\boldsymbol{s}}_{i_{\bar{\boldsymbol{x}}_i}}, \bar{\boldsymbol{X}}_{i-1} \neq \bar{\boldsymbol{x}}_{i-1})$$

$$= P(\bar{\boldsymbol{s}}_{i+1}, \bar{\boldsymbol{x}}_i) + P^M(\bar{\boldsymbol{s}}_{i_{\bar{\boldsymbol{x}}_i}}, \bar{\boldsymbol{x}}_{i-1}) - P^M(\bar{\boldsymbol{s}}_{i_{\bar{\boldsymbol{x}}_i}}, \bar{\boldsymbol{x}}_i) + P^M(\bar{\boldsymbol{s}}_{i_{\bar{\boldsymbol{x}}_i}}) - P^M(\bar{\boldsymbol{s}}_{i_{\bar{\boldsymbol{x}}_i}}, \bar{\boldsymbol{x}}_{i-1})$$

$$= P^M(\bar{\boldsymbol{s}}_{i_{\bar{\boldsymbol{x}}_i}}) + P(\bar{\boldsymbol{s}}_{i+1}, \bar{\boldsymbol{x}}_i) - P^M(\bar{\boldsymbol{s}}_{i_{\bar{\boldsymbol{x}}_i}}, \bar{\boldsymbol{x}}_i)$$

175    Again, by the composition axiom and $M \in \mathcal{M}_{\text{OBS}}$, $P^M(\bar{\boldsymbol{s}}_{i_{\bar{\boldsymbol{x}}_i}}, \bar{\boldsymbol{x}}_i) = P(\bar{\boldsymbol{s}}_i, \bar{\boldsymbol{x}}_i)$. Since $P^M(\bar{\boldsymbol{s}}_{i_{\bar{\boldsymbol{x}}_i}}) =$
176    $P^M_{\bar{\boldsymbol{x}}_i}(\bar{\boldsymbol{s}}_i)$, we have

$$P^M_{\bar{\boldsymbol{x}}_i}(\bar{\boldsymbol{s}}_{i+1}) = P^M_{\bar{\boldsymbol{x}}_i}(\bar{\boldsymbol{s}}_i) + P(\bar{\boldsymbol{s}}_{i+1}, \bar{\boldsymbol{x}}_i) - P(\bar{\boldsymbol{s}}_i, \bar{\boldsymbol{x}}_i).$$

177    Rearranging the above equation proves Eq. (36). Following a similar induction procedure in the proof
178    of Lem. 4, we have that for any $M \in \{M_1, M_2\}$,

$$P^M_{\bar{\boldsymbol{x}}_{k-1}}(\bar{\boldsymbol{s}}_k) = \Gamma(\bar{\boldsymbol{s}}_k, \bar{\boldsymbol{x}}_{k-1}). \tag{38}$$

179    As for $M_1$, for any $\bar{\boldsymbol{x}}_k^{k+1} \neq \bar{\boldsymbol{x}}_k$, we define

$$P^{M_1}(s_{k+1_{\bar{\boldsymbol{x}}_k^{k+1}}} | \bar{\boldsymbol{s}}_{k_{\bar{\boldsymbol{x}}_{k-1}}}, \bar{\boldsymbol{x}}_k) = 0$$

180    This implies

$$P^{M_1}_{\bar{\boldsymbol{x}}_k}(\bar{\boldsymbol{s}}_{k+1}) = P^{M_1}(\bar{\boldsymbol{s}}_{k+1_{\bar{\boldsymbol{x}}_k}}, \bar{\boldsymbol{x}}_k) + \sum_{\bar{\boldsymbol{x}}'_k \neq \bar{\boldsymbol{x}}_k} P^{M_1}(s_{k+1_{\bar{\boldsymbol{x}}_k}} | \bar{\boldsymbol{s}}_{k_{\bar{\boldsymbol{x}}_{k-1}}}, \bar{\boldsymbol{x}}'_k) P^{M_1}(\bar{\boldsymbol{s}}_{k_{\bar{\boldsymbol{x}}_{k-1}}}, \bar{\boldsymbol{x}}'_k)$$

$$= P^{M_1}(\bar{\boldsymbol{s}}_{k+1_{\bar{\boldsymbol{x}}_k}}, \bar{\boldsymbol{x}}_k).$$

181    By the composition axiom and $M_1 \in \mathcal{M}_{\text{OBS}}$, $P^{M_1}(\bar{\boldsymbol{s}}_{k+1_{\bar{\boldsymbol{x}}_k}}, \bar{\boldsymbol{x}}_k) = P(\bar{\boldsymbol{s}}_{k+1}, \bar{\boldsymbol{x}}_k)$, which gives

$$P^{M_1}_{\bar{\boldsymbol{x}}_k}(\bar{\boldsymbol{s}}_{k+1}) = P(\bar{\boldsymbol{s}}_{k+1}, \bar{\boldsymbol{x}}_k).$$

182    The above equation, together with Eq. (38), gives:

$$P^{M_1}_{\bar{\boldsymbol{x}}_k}(s_{k+1} | \bar{\boldsymbol{s}}_k) = \frac{P^{M_1}_{\bar{\boldsymbol{x}}_k}(\bar{\boldsymbol{s}}_{k+1})}{P^M_{\bar{\boldsymbol{x}}_{k-1}}(\bar{\boldsymbol{s}}_k)} = \frac{P(\bar{\boldsymbol{s}}_{k+1}, \bar{\boldsymbol{x}}_k)}{\Gamma(\bar{\boldsymbol{s}}_k, \bar{\boldsymbol{x}}_{k-1})} = a_{\bar{\boldsymbol{x}}_k, \bar{\boldsymbol{s}}_k}(s_{k+1}).$$

183    As for $M_2$, for any $\bar{\boldsymbol{x}}_k^{k+1} \neq \bar{\boldsymbol{x}}_k$, we define

$$P^{M_2}(s_{k+1_{\bar{\boldsymbol{x}}_k^{k+1}}} | \bar{\boldsymbol{s}}_{k_{\bar{\boldsymbol{x}}_{k-1}}}, \bar{\boldsymbol{x}}_k) = 1.$$

184    Following a similar procedure for proving Eq. (38), we have

$$P^M_{\bar{\boldsymbol{x}}_k}(\bar{\boldsymbol{s}}_{k+1}) = \Gamma(\bar{\boldsymbol{s}}_{k+1}, \bar{\boldsymbol{x}}_k).$$

185    Thus,

$$P^{M_2}_{\bar{\boldsymbol{x}}_k}(s_{k+1} | \bar{\boldsymbol{s}}_k) = \frac{P^{M_2}_{\bar{\boldsymbol{x}}_k}(\bar{\boldsymbol{s}}_{k+1})}{P^{M_2}_{\bar{\boldsymbol{x}}_{k-1}}(\bar{\boldsymbol{s}}_k)} = \frac{\Gamma(\bar{\boldsymbol{s}}_{k+1}, \bar{\boldsymbol{x}}_k)}{\Gamma(\bar{\boldsymbol{s}}_k, \bar{\boldsymbol{x}}_{k-1})} = b_{\bar{\boldsymbol{x}}_k, \bar{\boldsymbol{s}}_k}(s_{k+1}). \qquad \square$$

186 **Corollary 2.** *For a DTR, given* $P(\bar{\boldsymbol{s}}_K, \bar{\boldsymbol{x}}_K, y) > 0$,

$$\frac{E[Y|\bar{\boldsymbol{s}}_K, \bar{\boldsymbol{x}}_K]P(\bar{\boldsymbol{s}}_K, \bar{\boldsymbol{x}}_K)}{\Gamma(\bar{\boldsymbol{s}}_K, \bar{\boldsymbol{x}}_{K-1})} \le E_{\bar{\boldsymbol{x}}_K}[Y|\bar{\boldsymbol{s}}_k] \le 1 + \frac{(E[Y|\bar{\boldsymbol{s}}_K, \bar{\boldsymbol{x}}_K] - 1)P(\bar{\boldsymbol{s}}_K, \bar{\boldsymbol{x}}_K)}{\Gamma(\bar{\boldsymbol{s}}_K, \bar{\boldsymbol{x}}_{K-1})}.$$

187 *Proof.* By basic probabilistic operations,

$$E_{\bar{\boldsymbol{x}}_K}[Y|\bar{\boldsymbol{s}}_k] = \frac{E_{\bar{\boldsymbol{x}}_K}[Y|\bar{\boldsymbol{s}}_K]P_{\bar{\boldsymbol{x}}_K}(\bar{\boldsymbol{s}}_K)}{P_{\bar{\boldsymbol{x}}_K}(\bar{\boldsymbol{s}}_K)}.$$

188 Note the counterfactual $Y_{\bar{\boldsymbol{x}}_K, \bar{\boldsymbol{s}}_K}(\boldsymbol{u}) \in [0, 1]$. Following a similar argument as Lem. 1,

$$E_{\bar{\boldsymbol{x}}_K}[Y|\bar{\boldsymbol{s}}_K]P_{\bar{\boldsymbol{x}}_K}(\bar{\boldsymbol{s}}_K) - P_{\bar{\boldsymbol{x}}_K}(\bar{\boldsymbol{s}}_K) \le E[Y|\bar{\boldsymbol{s}}_K, \bar{\boldsymbol{x}}_K]P(\bar{\boldsymbol{s}}_K, \bar{\boldsymbol{x}}_K) - P(\bar{\boldsymbol{s}}_K, \bar{\boldsymbol{x}}_K).$$

189 This implies

$$E_{\bar{\boldsymbol{x}}_K}[Y|\bar{\boldsymbol{s}}_k] \le 1 + \frac{(E[Y|\bar{\boldsymbol{s}}_K, \bar{\boldsymbol{x}}_K] - 1)P(\bar{\boldsymbol{s}}_K, \bar{\boldsymbol{x}}_K)}{P_{\bar{\boldsymbol{x}}_K}(\bar{\boldsymbol{s}}_K)}$$

190 Since $E[Y|\bar{\boldsymbol{s}}_K, \bar{\boldsymbol{x}}_K] \le 1$, $E_{\bar{\boldsymbol{x}}_K}[Y|\bar{\boldsymbol{s}}_k]$ is upper-bounded when $P_{\bar{\boldsymbol{x}}_K}(\bar{\boldsymbol{s}}_K)$ is the maximal. Since $\bar{\boldsymbol{S}}_K$
191 are non-descendants of $X_K$, $P_{\bar{\boldsymbol{x}}_K}(\bar{\boldsymbol{s}}_K) = P_{\bar{\boldsymbol{x}}_{K-1}}(\bar{\boldsymbol{s}}_K)$. By Lem. 4,

$$E_{\bar{\boldsymbol{x}}_K}[Y|\bar{\boldsymbol{s}}_k] \le 1 + \frac{(E[Y|\bar{\boldsymbol{s}}_K, \bar{\boldsymbol{x}}_K] - 1)P(\bar{\boldsymbol{s}}_K, \bar{\boldsymbol{x}}_K)}{\Gamma(\bar{\boldsymbol{s}}_k, \bar{\boldsymbol{x}}_{k-1})}.$$

192 By definition, $P_{\bar{\boldsymbol{x}}_K}(y, \bar{\boldsymbol{s}}_K) = P(y_{\bar{\boldsymbol{x}}_K}, \bar{\boldsymbol{s}}_{K_{\bar{\boldsymbol{x}}_{K-1}}})$. By basic probabilistic operations,

$$E_{\bar{\boldsymbol{x}}_K}[Y|\bar{\boldsymbol{s}}_k] \ge \frac{E[Y_{\bar{\boldsymbol{x}}_K}|\bar{\boldsymbol{s}}_{K_{\bar{\boldsymbol{x}}_{K-1}}}, \bar{\boldsymbol{x}}_K]P(\bar{\boldsymbol{s}}_{K_{\bar{\boldsymbol{x}}_{K-1}}}, \bar{\boldsymbol{x}}_K)}{P_{\bar{\boldsymbol{x}}_{K-1}}(\bar{\boldsymbol{s}}_K)}.$$

193 By the composition axiom, $\bar{\boldsymbol{S}}_{K_{\bar{\boldsymbol{x}}_{K-1}}} = \bar{\boldsymbol{S}}_{K-1}, Y_{\bar{\boldsymbol{x}}_K} = Y$ if $\bar{\boldsymbol{X}}_K = \bar{\boldsymbol{x}}_K$. Applying Lem. 4 gives

$$E_{\bar{\boldsymbol{x}}_K}[Y|\bar{\boldsymbol{s}}_k] \ge \frac{E[Y|\bar{\boldsymbol{s}}_K, \bar{\boldsymbol{x}}_K]P(\bar{\boldsymbol{s}}_K, \bar{\boldsymbol{x}}_K)}{P_{\bar{\boldsymbol{x}}_K}(\bar{\boldsymbol{s}}_K)} = \frac{E[Y|\bar{\boldsymbol{s}}_K, \bar{\boldsymbol{x}}_K]P(\bar{\boldsymbol{s}}_K, \bar{\boldsymbol{x}}_K)}{\Gamma(\bar{\boldsymbol{s}}_K, \bar{\boldsymbol{x}}_{K-1})}. \qquad \square$$

194 **Proof of Theorems 7 and 8**

195 **Lemma 5.** *Fix* $\epsilon > 0$, $\delta \in (0, 1)$. *With probability (w.p.) of at least* $1 - \delta$, *it holds for any* $T > 1$,
196 $R_\epsilon(T)$ *of* `UC-DTR` *with parameter* $\delta$ *and causal bounds* $\mathcal{C}$ *is bounded by*

$$R_\epsilon(T) \le \min\left\{ 12K\sqrt{|\boldsymbol{\mathcal{S}}||\boldsymbol{\mathcal{X}}|T_\epsilon \log(2K|\boldsymbol{\mathcal{S}}||\boldsymbol{\mathcal{X}}|T/\delta)}, \|\mathcal{C}\|_1 T_\epsilon \right\} + 4K\sqrt{T_\epsilon \log(2T/\delta)}$$

197 *Proof.* Note that causal bounds $\mathcal{C}$ is a set $\{\mathcal{C}_1, \dots, \mathcal{C}_K\}$ where for $k = 1, \dots, K-1$,

$$\mathcal{C}_k = \left\{ \forall \bar{\boldsymbol{s}}_{k+1}, \bar{\boldsymbol{x}}_k : \left[a_{\bar{\boldsymbol{x}}_k, \bar{\boldsymbol{s}}_k}(s_{k+1}), b_{\bar{\boldsymbol{x}}_k, \bar{\boldsymbol{s}}_k}(s_{k+1})\right] \right\},$$

$$\text{and } \mathcal{C}_K = \left\{ \forall \bar{\boldsymbol{s}}_K, \bar{\boldsymbol{x}}_K : \left[a_{\bar{\boldsymbol{x}}_K, \bar{\boldsymbol{s}}_K}, b_{\bar{\boldsymbol{x}}_K, \bar{\boldsymbol{s}}_K}\right] \right\}. \tag{39}$$

198 $\mathcal{M}^c$ is a set of DTRs such that for any $M \in \mathcal{M}^c$, its causal quantities $P_{\bar{\boldsymbol{x}}_k}(s_{k+1}|\bar{\boldsymbol{s}}_k)$ and $E_{\bar{\boldsymbol{x}}_K}[Y|\bar{\boldsymbol{s}}_K]$
199 satisfy the causal bounds $\mathcal{C}$, i.e.,

$$P_{\bar{\boldsymbol{x}}_k}(s_{k+1}|\bar{\boldsymbol{s}}_k) \in \left[a_{\bar{\boldsymbol{x}}_k, \bar{\boldsymbol{s}}_k}(s_{k+1}), b_{\bar{\boldsymbol{x}}_k, \bar{\boldsymbol{s}}_k}(s_{k+1})\right], \quad \text{and } E_{\bar{\boldsymbol{x}}_K}[Y|\bar{\boldsymbol{s}}_K] \in \left[a_{\bar{\boldsymbol{x}}_K, \bar{\boldsymbol{s}}_K}, b_{\bar{\boldsymbol{x}}_K, \bar{\boldsymbol{s}}_K}\right]. \tag{40}$$

200 Let $\mathcal{M}_t^c = \mathcal{M}_t \cap \mathcal{M}^c$. Since $\mathcal{M}_t^c \subseteq \mathcal{M}_t$, following a similar argument in [4, C.1], we have

$$P(M^* \in \mathcal{M}_t^c) \le P(M^* \in \mathcal{M}_t) \le \frac{\delta}{4t^2}. \tag{41}$$

201 Since $\sum_{t=1}^{\infty} \frac{1}{4t^2} \le \frac{\pi^2}{24}\delta < \frac{\delta}{2}$, it follows that with probability at least $1 - \frac{\delta}{2}$, $M^* \in \mathcal{M}_c^t$ for all
202 episodes $t = 1, 2, \dots$.

203 Following the proof of Lem. 3, we have

$$R_\epsilon(T) \le K\sqrt{6T_\epsilon \log(2T/\delta)} + \sqrt{\frac{3T_\epsilon \log(2T/\delta)}{2}}$$

$$+ \sum_{k=1}^{K-1} \sum_{t \in L_\epsilon} V_{\boldsymbol{\pi}_t}(\bar{\boldsymbol{S}}_k^t, \bar{\boldsymbol{X}}_k^t; M_t^{(k)}) - V_{\boldsymbol{\pi}_t}(\bar{\boldsymbol{S}}_k^t, \bar{\boldsymbol{X}}_k^t; M_t^{(k+1)}) \tag{42}$$

$$+ \sum_{t \in L_\epsilon} \left(E_{\bar{\boldsymbol{X}}_K^t}^{M_t}[Y|\bar{\boldsymbol{S}}_K^t] - E_{\bar{\boldsymbol{X}}_K^t}[Y|\bar{\boldsymbol{S}}_K^t]\right). \tag{43}$$

204 It thus suffices to bound quantities in Eqs. (42) and (43) separately.

205 **Bounding Eq. (42)** By Eq. (22) and basic probabilistic operations,

$$
\begin{aligned}
& V_{\boldsymbol{\pi}_t}(\bar{\boldsymbol{S}}_k^t, \bar{\boldsymbol{X}}_k^t; M_t^{(k)}) - V_{\boldsymbol{\pi}_t}(\bar{\boldsymbol{S}}_k^t, \bar{\boldsymbol{X}}_k^t; M_t^{(k+1)}) \\
& = \sum_{s_{k+1}} (P^{M_t}(s_{k+1}|\bar{\boldsymbol{S}}_k, \bar{\boldsymbol{X}}_k) - P(s_{k+1}|\bar{\boldsymbol{S}}_k, \bar{\boldsymbol{X}}_k)) V_{\boldsymbol{\pi}_t}(s_{k+1}, \bar{\boldsymbol{S}}_k^t, \bar{\boldsymbol{X}}_k^t; M_t) \\
& \leq \left\| P_{\bar{\boldsymbol{x}}_k}^{M_t}(\cdot|\bar{\boldsymbol{s}}_k) - P_{\bar{\boldsymbol{x}}_k}(\cdot|\bar{\boldsymbol{s}}_k) \right\|_1 \max_{s_{k+1}} V_{\boldsymbol{\pi}_t}(s_{k+1}, \bar{\boldsymbol{S}}_k^t, \bar{\boldsymbol{X}}_k^t; M_t) \\
& \leq \min \left\{ 2\sqrt{6|\mathcal{S}_{k+1}|\log(2K|\bar{\boldsymbol{\mathcal{S}}}_k||\bar{\boldsymbol{\mathcal{X}}}_k|T/\delta)} \frac{1}{\sqrt{\max\{1, N^t(\bar{\boldsymbol{S}}_k^t, \bar{\boldsymbol{X}}_k^t)\}}}, \|\mathcal{C}_k\|_1 \right\}
\end{aligned}
$$

206 The last step follows from Eqs. (16) and (40). We thus have

$$
\begin{aligned}
& \sum_{t\in L_\epsilon} V_{\boldsymbol{\pi}_t}(\bar{\boldsymbol{S}}_k^t, \bar{\boldsymbol{X}}_k^t; M_t^{(k)}) - V_{\boldsymbol{\pi}_t}(\bar{\boldsymbol{S}}_k^t, \bar{\boldsymbol{X}}_k^t; M_t^{(k+1)}) \\
& \leq \sum_{t\in L_\epsilon} \min \left\{ 2\sqrt{6|\mathcal{S}_{k+1}|\log(2K|\bar{\boldsymbol{\mathcal{S}}}_k||\bar{\boldsymbol{\mathcal{X}}}_k|T/\delta)} \frac{1}{\sqrt{\max\{1, N^t(\bar{\boldsymbol{S}}_k^t, \bar{\boldsymbol{X}}_k^t)\}}}, \|\mathcal{C}_k\|_1 \right\} \\
& \leq \min \left\{ \sum_{t\in L_\epsilon} 2\sqrt{6|\mathcal{S}_{k+1}|\log(2K|\bar{\boldsymbol{\mathcal{S}}}_k||\bar{\boldsymbol{\mathcal{X}}}_k|T/\delta)} \frac{1}{\sqrt{\max\{1, N^t(\bar{\boldsymbol{S}}_k^t, \bar{\boldsymbol{X}}_k^t)\}}}, \sum_{t\in L_\epsilon} \|\mathcal{C}_k\|_1 \right\} \\
& \leq \min \left\{ 2(\sqrt{2}+1)\sqrt{6T_\epsilon|\bar{\boldsymbol{\mathcal{S}}}_{k+1}||\bar{\boldsymbol{\mathcal{X}}}_k|\log(2K|\bar{\boldsymbol{\mathcal{S}}}_k||\bar{\boldsymbol{\mathcal{X}}}_k|T/\delta)}, \|\mathcal{C}_k\|_1 T_\epsilon \right\}
\end{aligned}
$$

207 The last step follows from results in [4, D] and $|L_\epsilon| = T_\epsilon$. Eq. (42) could thus be written as:

$$
\begin{aligned}
& \sum_{k=1}^{K-1} \sum_{t\in L_\epsilon} V_{\boldsymbol{\pi}_t}(\bar{\boldsymbol{S}}_k^t, \bar{\boldsymbol{X}}_k^t; M_t^{(k)}) - V_{\boldsymbol{\pi}_t}(\bar{\boldsymbol{S}}_k^t, \bar{\boldsymbol{X}}_k^t; M_t^{(k+1)}) \\
& \leq \sum_{k=1}^{K-1} \min \left\{ 2(\sqrt{2}+1)\sqrt{6T_\epsilon|\bar{\boldsymbol{\mathcal{S}}}_{k+1}||\bar{\boldsymbol{\mathcal{X}}}_k|\log(2K|\bar{\boldsymbol{\mathcal{S}}}_k||\bar{\boldsymbol{\mathcal{X}}}_k|T/\delta)}, \|\mathcal{C}_k\|_1 T_\epsilon \right\} \\
& \leq \min \left\{ \sum_{k=1}^{K-1} 2(\sqrt{2}+1)\sqrt{6T_\epsilon|\bar{\boldsymbol{\mathcal{S}}}_{k+1}||\bar{\boldsymbol{\mathcal{X}}}_k|\log(2K|\bar{\boldsymbol{\mathcal{S}}}_k||\bar{\boldsymbol{\mathcal{X}}}_k|T/\delta)}, \sum_{k=1}^{K-1} \|\mathcal{C}_k\|_1 T_\epsilon \right\}
\end{aligned}
$$

208 Thus,

$$
\begin{aligned}
& \sum_{k=1}^{K-1} \sum_{t\in L_\epsilon} V_{\boldsymbol{\pi}_t}(\bar{\boldsymbol{S}}_k^t, \bar{\boldsymbol{X}}_k^t; M_t^{(k)}) - V_{\boldsymbol{\pi}_t}(\bar{\boldsymbol{S}}_k^t, \bar{\boldsymbol{X}}_k^t; M_t^{(k+1)}) \\
& \leq \min \left\{ (K-1)2(\sqrt{2}+1)\sqrt{6T_\epsilon|\bar{\boldsymbol{\mathcal{S}}}||\bar{\boldsymbol{\mathcal{X}}}|\log(2K|\bar{\boldsymbol{\mathcal{S}}}||\bar{\boldsymbol{\mathcal{X}}}|T/\delta)}, \sum_{k=1}^{K-1} \|\mathcal{C}_k\|_1 T_\epsilon \right\}.
\end{aligned} \tag{44}
$$

209 **Bounding Eq. (43)** Since both $M^*, M_t$ are in the set $\mathcal{M}_t^c$,

$$
\begin{aligned}
E_{\bar{\boldsymbol{X}}_K^t}^{M_t}[Y|\bar{\boldsymbol{S}}_K^t] - E_{\bar{\boldsymbol{X}}_K^t}[Y|\bar{\boldsymbol{S}}_K^t] & \leq \left| E_{\bar{\boldsymbol{x}}_K}^{M_t}[Y|\bar{\boldsymbol{s}}_K] - \hat{E}_{\bar{\boldsymbol{x}}_K}^t[Y|\bar{\boldsymbol{s}}_K] \right| + \left| E_{\bar{\boldsymbol{x}}_K^t}[Y|\bar{\boldsymbol{S}}_K^t] - \hat{E}_{\bar{\boldsymbol{x}}_K}^t[Y|\bar{\boldsymbol{s}}_K] \right| \\
& \leq \min \left\{ 2\sqrt{2\log(2K|\boldsymbol{\mathcal{S}}||\boldsymbol{\mathcal{X}}|T/\delta)} \frac{1}{\sqrt{\max\{1, N^t(\bar{\boldsymbol{S}}_K^t, \bar{\boldsymbol{X}}_K^t)\}}}, \|\mathcal{C}_K\|_1 \right\}
\end{aligned}
$$

210 Eq. (43) can thus be written as:

$$
\begin{aligned}
& \sum_{t\in L_\epsilon} \left( E_{\bar{\boldsymbol{X}}_K^t}^{M_t}[Y|\bar{\boldsymbol{S}}_K^t] - E_{\bar{\boldsymbol{X}}_K^t}[Y|\bar{\boldsymbol{S}}_K^t] \right) \\
& \leq \sum_{t\in L_\epsilon} \min \left\{ 2\sqrt{2\log(2K|\boldsymbol{\mathcal{S}}||\boldsymbol{\mathcal{X}}|T/\delta)} \frac{1}{\sqrt{\max\{1, N^t(\bar{\boldsymbol{S}}_K^t, \bar{\boldsymbol{X}}_K^t)\}}}, \|\mathcal{C}_K\|_1 \right\} \\
& \leq \min \left\{ \sum_{t\in L_\epsilon} 2\sqrt{2\log(2K|\boldsymbol{\mathcal{S}}||\boldsymbol{\mathcal{X}}|T/\delta)} \frac{1}{\sqrt{\max\{1, N^t(\bar{\boldsymbol{S}}_K^t, \bar{\boldsymbol{X}}_K^t)\}}}, \sum_{t\in L_\epsilon} \|\mathcal{C}_K\|_1 \right\}.
\end{aligned}
$$

The last step follows from Eqs. (17) and (40). From results in [4, D], we have

$$
\sum_{t \in L_\epsilon} \left( E^{M_t}_{\bar{\boldsymbol{X}}^t_K}[Y|\bar{\boldsymbol{S}}^t_K] - E_{\bar{\boldsymbol{X}}^t_K}[Y|\bar{\boldsymbol{S}}^t_K] \right)
$$
$$
\leq \min\left\{ 2(\sqrt{2}+1)\sqrt{2T_\epsilon|\bar{\boldsymbol{S}}||\bar{\boldsymbol{\mathcal{X}}}|\log(2K|\bar{\boldsymbol{S}}||\bar{\boldsymbol{\mathcal{X}}}|T/\delta)}, \left\|\boldsymbol{\mathcal{C}}_K\right\|_1 T_\epsilon \right\}. \tag{45}
$$

Eqs. (44) and (45) together give:

$$
R_\epsilon(T) \leq K\sqrt{6T_\epsilon\log(2T/\delta)} + \sqrt{\frac{3T_\epsilon\log(2T/\delta)}{2}}
$$
$$
+ \min\left\{(K-1)2(\sqrt{2}+1)\sqrt{6T_\epsilon|\bar{\boldsymbol{S}}||\bar{\boldsymbol{\mathcal{X}}}|\log(2K|\bar{\boldsymbol{S}}||\bar{\boldsymbol{\mathcal{X}}}|T/\delta)}, \sum_{k=1}^{K-1}\left\|\boldsymbol{\mathcal{C}}_k\right\|_1 T_\epsilon\right\} \tag{46}
$$
$$
+ \min\left\{2(\sqrt{2}+1)\sqrt{2T_\epsilon|\bar{\boldsymbol{S}}||\bar{\boldsymbol{\mathcal{X}}}|\log(2K|\bar{\boldsymbol{S}}||\bar{\boldsymbol{\mathcal{X}}}|T/\delta)}, \left\|\boldsymbol{\mathcal{C}}_K\right\|_1 T_\epsilon\right\}.
$$

A quick simplification gives:

$$
R_\epsilon(T) \leq \min\left\{12K\sqrt{|\boldsymbol{S}||\boldsymbol{\mathcal{X}}|T_\epsilon\log(2K|\boldsymbol{S}||\boldsymbol{\mathcal{X}}|T/\delta)}, \left\|\boldsymbol{\mathcal{C}}\right\|_1 T_\epsilon\right\} + 4K\sqrt{T_\epsilon\log(2T/\delta)}. \qquad \square
$$

**Theorem 7.** *Fix a $\delta \in (0,1)$. With probability of at least $1-\delta$, it holds for any $T > 1$, the regret of* UC$^c$-DTR *with parameter $\delta$ and causal bounds $\boldsymbol{\mathcal{C}}$ is bounded by*

$$
R(T) \leq \min\left\{12K\sqrt{|\boldsymbol{S}||\boldsymbol{\mathcal{X}}|T\log(2K|\boldsymbol{S}||\boldsymbol{\mathcal{X}}|T/\delta)}, \left\|\boldsymbol{\mathcal{C}}\right\|_1 T\right\} + 4K\sqrt{T\log(2T/\delta)}.
$$

*Proof.* Fix $\epsilon = 0$. Naturally, $T_\epsilon = T$ and $R_\epsilon(T) = R(T)$. By Lem. 5,

$$
R(T) \leq \min\left\{12K\sqrt{|\boldsymbol{S}||\boldsymbol{\mathcal{X}}|T\log(2K|\boldsymbol{S}||\boldsymbol{\mathcal{X}}|T/\delta)}, \left\|\boldsymbol{\mathcal{C}}\right\|_1 T\right\} + 4K\sqrt{T\log(2T/\delta)}. \qquad \square
$$

**Theorem 8.** *For any $T \geq 1$, with parameter $\delta = \frac{1}{T}$ and causal bounds $\boldsymbol{\mathcal{C}}$, the expected regret of* UC$^c$-DTR *is bounded by*

$$
E[R(T)] \leq \max_{\boldsymbol{\pi} \in \boldsymbol{\Pi}_{\boldsymbol{\mathcal{C}}}^-}\left\{\frac{33^2 K^2|\boldsymbol{S}||\boldsymbol{\mathcal{X}}|\log(T)}{\Delta_{\boldsymbol{\pi}}} + \frac{32}{\Delta_{\boldsymbol{\pi}}^3} + \frac{4}{\Delta_{\boldsymbol{\pi}}}\right\} + 1.
$$

*Proof.* Let $\tilde{R}_\epsilon(T)$ denote the regret cumulated in $\epsilon$-good episode up to $T$ steps. By Eqs. (41) and (46),

$$
E[R(T)] \leq E[R_\epsilon(T)I_{M^* \in \mathcal{M}_t^c}] + E[\tilde{R}_\epsilon(T)I_{M^* \in \mathcal{M}_t^c}] + \sum_{t=1}^{T} P(M \notin \mathcal{M}_t^c)
$$
$$
\leq \min\left\{12K\sqrt{|\boldsymbol{S}||\boldsymbol{\mathcal{X}}|T\log(2K|\boldsymbol{S}||\boldsymbol{\mathcal{X}}|T/\delta)}, \left\|\boldsymbol{\mathcal{C}}\right\|_1 T\right\} + 4K\sqrt{T\log(2T/\delta)}
$$
$$
+ E[\tilde{R}_\epsilon(T)I_{M^* \in \mathcal{M}_t^c}] + \frac{\delta}{T}
$$
$$
\leq 23K\sqrt{|\boldsymbol{S}||\boldsymbol{\mathcal{X}}|T_\epsilon\log(T/\delta)} + E[\tilde{R}_\epsilon(T)I_{M^* \in \mathcal{M}_t^c}] + \frac{\delta}{T}
$$

Fix $\delta = \frac{1}{T}$, it is immediate from Eq. (28) that

$$
E[R(T)] \leq \frac{23^2 K^2|\boldsymbol{S}||\boldsymbol{\mathcal{X}}|\log(T^2)}{\epsilon} + E[\tilde{R}_\epsilon(T)I_{M^* \in \mathcal{M}_t^c}] + 1. \tag{47}
$$

Note that when $M^* \in \mathcal{M}_t^c$, the maximal expected reward of any $\boldsymbol{\pi}_t$ over all instances in the family of DTRs $\mathcal{M}_t^c$ must be no less than the true optimal value $V_{\boldsymbol{\pi}^*}(M^*)$. In words, $\boldsymbol{\Pi}_{\boldsymbol{C}}^-$ is the effective policy space of UC$^c$-DTR procedure. Let $\Delta = \arg\min_{\boldsymbol{\pi} \in \boldsymbol{\Pi}_{\boldsymbol{C}}^-}\Delta_{\boldsymbol{\pi}}$. Fix $\epsilon = \frac{\Delta}{2}$, Eq. (47) implies:

$$
E[R(T)] \leq \frac{33^2 K^2|\boldsymbol{S}||\boldsymbol{\mathcal{X}}|\log(T)}{\Delta} + E[\tilde{R}_{\frac{\Delta}{2}}(T)I_{M^* \in \mathcal{M}_t^c}] + 1.
$$

Among quantities in the above equation, $E[\tilde{R}_{\frac{\Delta}{2}}(T)I_{M^* \in \mathcal{M}_t^c}]$ can be bounded following a similar procedure in the proof of Thm. 2, which proves the statement. $\qquad \square$

## Appendix II. Estimation of Causal Bounds

The bounds developed in the main text are functions of the observational distribution $P(\bar{\boldsymbol{s}}_K, \bar{\boldsymbol{x}}_K, y)$ which is identifiable by the sampling process, and so can be estimated consistently. Bounding causal effects from a finite set of observations is more involved, due to the issues of sampling variability. We now present efficient methods to address these issues.

Given a finite set of observational samples $\{\bar{\boldsymbol{S}}_K^i, \bar{\boldsymbol{X}}_K^i, Y^i\}_{i=1}^n$, let $\hat{P}(\bar{\boldsymbol{s}}_K, \bar{\boldsymbol{x}}_K)$ denote the sample mean estimate of $P(\bar{\boldsymbol{s}}_K, \bar{\boldsymbol{x}}_K)$. Fix $\delta \in (0, 1)$. W.p. at least $1 - \delta$, the L1-deviation of the true distribution $P(\bar{\boldsymbol{s}}_K, \bar{\boldsymbol{x}}_K)$ and the empirical distribution $\hat{P}(\bar{\boldsymbol{s}}_K, \bar{\boldsymbol{x}}_K)$ over state-action domains $\boldsymbol{\mathcal{S}} \times \boldsymbol{\mathcal{X}}$ from $n$ samples is bounded according to [9] by

$$\left\| P(\cdot) - \hat{P}(\cdot) \right\|_1 \leq \sqrt{2|\boldsymbol{\mathcal{S}}||\boldsymbol{\mathcal{X}}|\log(2/\delta)/n}. \tag{48}$$

We could derive confidence bounds of probabilities $P_{\bar{\boldsymbol{x}}_k}(s_{k+1}|\bar{\boldsymbol{s}}_k)$ for all $k = 1, \ldots, K-1$ w.p. $1 - \delta$ by optimizing the causal bounds $\left[ a_{\bar{\boldsymbol{x}}_k, \bar{\boldsymbol{s}}_k}(s_{k+1}), b_{\bar{\boldsymbol{x}}_k, \bar{\boldsymbol{s}}_k}(s_{k+1}) \right]$ subject to convex polytope defined in Eq. (48) and probabilistic constraints $P(\bar{\boldsymbol{s}}_K, \bar{\boldsymbol{x}}_K) \in [0, 1]$ and $\sum_{\bar{\boldsymbol{s}}_K, \bar{\boldsymbol{x}}_K} P(\bar{\boldsymbol{s}}_K, \bar{\boldsymbol{x}}_K) = 1$. The objective functions in Eq. (9) are ratios of linear functions, leading to a linear-fractional program (LFP). A LFP can be transformed into an equivalent linear program (LP) by [2], which is solvable using standard LP algorithms. The expected reward $E_{\bar{\boldsymbol{x}}_K}[Y|\bar{\boldsymbol{s}}_K]$ could be bounded following a similar procedure.

## Appendix III. Experimental Setup

In this section, we provide details about the setup of experiments in the main text. For all experiments, we test sequentially randomized trials (*rand*), UC-DTR algorithm (*uc-dtr*) and the causal UC-DTR (*$uc^c$-dtr*) with causal bounds derived from $1 \times 10^5$ observational samples. Each experiment lasts for $T = 1.1 \times 10^4$ episodes. The parameter $\delta = 1/KT$ for *uc-dtr* and *$uc^c$-dtr* where $K$ is the total stages of interventions. For all algorithms, we measure their cumulative regret over 200 repetitions.

**Random DTRs**  We generate 200 instances of the counterfactual DTR defined in Def. 1. We assume treatments $X_1, X_2$, states $S_1, S_2$ and primary outcome $Y$ are all binary variable. The probabilities of the counterfactual distribution $P(s_1, x_1, s_{2_{x_1}}, x_{2_{x_1}}, y_{\bar{\boldsymbol{x}}_2})$ are drawn uniformly at random over $[0, 1]$.

**Cancer Treatment**  We test the survival model of patients inspired by the two-stage clinical trial conducted by the Cancer and Leukemia Group B [5, 8]. Protocol 8923 was a double-blind, placebo controlled two-stage trial reported by [7] examining the effects of infusions of granulocyte-macrophage colony-stimulating factor (GM-CSF) after initial chemotherapy. Patients were randomized initially to GM-CSF or placebo following standard chemotherapy. Later, patients meeting the criteria of complete remission were offered a second randomization to one of two intensification treatments.

We will describe this treatment procedure using the DTR with $K = 2$. $X_1, X_2 \in \{0, 1\}$ represent treatments; $S_1 = \emptyset$ and $S_2$ indicates the observed remission after the first treatment (0 stands for no remission and 1 for complete remission); $Y$ indicates the survival of patients at the time of recording. The exogenous variable $U$ is the age of patients where $U = 1$ if the patient is old and $U = 0$ otherwise. Values of $U$ are drawn from a distribution $P(u)$ where $P(U = 1) = 0.2358$. Values of $S_2$ are drawn from a distribution $P_{x_1}(s_2)$ described in Table 1.

|         | $X_1 = 0$ | $X_1 = 1$ |
|---------|-----------|-----------|
| $U = 0$ | 0.8101    | 0.0883    |
| $U = 1$ | 0.7665    | 0.2899    |

Table 1: Probabilities of the distribution $P(S_2 = 1|u, x_1)$.

Let $T_1, T_2$ denote the potential survival time induced by treatment $X_1, X_2$ respectively. Values of $T_1, T_2$ are decided by functions defined as follows:

$$T_1 \leftarrow \min\{(1 - S_2)T_1^* + S_2(T_2^* + T_3^*), L\}, \quad T_2 \leftarrow \min\{(1 - S_2)T_1^* + S_2(T_2^* + T_4^*), L\}$$

where $L = 1.5$. Let $exp(\beta)$ denote an exponential distribution with mean $1/\beta$. Values of $T_1^*, T_2^*, T_3^*$ are drawn from exponential distributions defined as follows:

$$T_1^* \sim exp(\beta_{u,x_1}^1), \qquad T_2^* \sim exp(\beta_{u,x_1}^2), \qquad T_3^* \sim exp(\beta_{u,x_1}^3)$$

Given $T_3^*$, values of $T_4^*$ are drawn from distribution

$$T_4^* \sim exp(\beta_{u,x_1}^3 + \beta_{u,x_1}^4 T_3^*).$$

The total survival time $T$ of a patient is decided as follows:

$$T \leftarrow (1 - S_2)T_1 + S_2(1 - X_2)T_1 + S_2 X_2 T_2.$$

The parameters $\boldsymbol{\beta}_{u,x_1} = (\beta_{u,x_1}^1, \beta_{u,x_1}^2, \beta_{u,x_1}^3, \beta_{u,x_1}^4)$ are described in Table 2.

|  |  | $\beta_{u,x_1}^1$ | $\beta_{u,x_1}^2$ | $\beta_{u,x_1}^3$ | $\beta_{u,x_1}^4$ |
|---|---|---|---|---|---|
| $U = 0$ | $X_1 = 0$ | 4.3063 | 4.9607 | 0.8737 | 4.2538 |
|  | $X_1 = 1$ | 0.8286 | 8.2074 | 8.7975 | 7.6468 |
| $U = 1$ | $X_1 = 0$ | 2.6989 | 0.0235 | 5.9835 | 6.8059 |
|  | $X_1 = 1$ | 3.6036 | 1.1007 | 9.4426 | 7.3960 |

Table 2: Parameters $\boldsymbol{\beta}_{u,x_1}$.

The primary outcome $Y$ is the survival of the patient at the time of observation $t = 1$. Values of $Y$ are decided by the indicator function $Y \leftarrow I_{T>1}$.

We generate the confounded observational data following a sequence of decision rules $X_1 \sim \pi_1(X_1|U), X_2 \sim \pi_2(X_2|U, X_1, S_2)$. The policy $\pi_1(X_1|U)$ is a conditional distribution mapping from $U$ to the domain of $X_1$ where $\pi_1(X_1 = 1|U = 0) = 0.5102$ and $\pi_1(X_1 = 1|U = 1) = 0.2433$. Similarly, $\pi_2(X_2|U, X_1, S_2)$ is a conditional distribution mapping from $U, X_1, S_2$ to the domain of $X_2$; Table 3 describes its parametrization.

|  | $X_1 = 0$ | | $X_1 = 1$ | |
|---|---|---|---|---|
|  | $S_2 = 0$ | $S_2 = 1$ | $S_2 = 0$ | $S_2 = 1$ |
| $U = 0$ | 0.2173 | 0.8696 | 0.6195 | 0.4641 |
| $U = 1$ | 0.8869 | 0.0103 | 0.5314 | 0.4339 |

Table 3: Probabilities of $\pi_2(X_2 = 1|U, X_1, S_2)$.