[Reviews · NeurIPS 2019]

Reviewer 1



# Originality Although some initial work [1, 2, 3] on combining causal inference and RL has been explored, this paper has another attempt to combine both from the perspective of DTR to learn a policy for an unknown DTR by leveraging the confounded observational data. It is a really novel and promising direction worth exploring further in the future. # Quality The claims presented in the paper are supported both in theory and in practice (i.e., simulated data and real-world data). But I have a question about the finite states. In Algorithm I, the defined event counts are required to be calculated over the states and treatments. What if the state/action dimension is very high? In this case, it would be difficult to obtain accurate estimates of the two terms in Step 3. Any idea? # Clarity The paper is well organised and is easy to follow. # Significance I believe this work would be of great interest to both RL and causal inference communities, and would potentially have a great number of applications in real world, in particular in healthcare/medicine, education, sociology, etc. As I mentioned previously, however, some shortcomings of the proposed algorithm (e.g., the state/action dimension) might limit its applicability in many real-world applications. [1] Causal Reasoning from Meta-reinforcement Learning, Dasgupta et al. 2019 [2] Deconfounding Reinforcement Learning in Observational Settings, Lu et al. 2018 [3] Learning Causal State Representations of Partially Observable Environments, Zhang et al. 2019

Reviewer 2



Update after rebuttal: Due to author comments and, in particular, discussions with the other reviewers, I have updated my score from 4 to a weak accept 6. For the future draft, aside from the revisions and clarifications the authors have promised in the rebuttal, I recommend the following (slight) modifications to improve the manuscript: The motivation in the introduction would be strengthened by drawing clearer connections to the real world. The authors should consider picking a specific real world example and illustrating the method through that example (even if it's not possible to provide simulation results on such an example). In line with this, the authors should be careful about discussion of safe-RL. Typically such methods involve use of constraints to ensure safety, but it does not appear the authors explicitly use or discuss such methods here. ------------------------------------------------- Detailed comments and questions: 1) The paper is wholly motivated through the lens of precision medicine and other healthcare contexts. The authors approach is primarily an online learning approach. In other words, the proposed algorithm will be taking actions on patients that will prove to be incorrect and could ultimately kill patients. Online learning is generally considered too unsafe for use in healthcare for this reason. Do the authors have evidence to back up the notion that this type of work would be adopted in a healthcare setting? At one point the authors seem to imply that doctors (as "agents") already follow this sort of behavior, however the decisions doctors make are 1) motivated by observations of the patients in question and 2) are backed up by substantial medical training. 2) "Most of the current work in the causal inference literature focus on trying to identify E_pi[Y] from finite observational data". Could the authors please provide examples of this work? The citations given from E.g. Pearl and collaborators is, for the most part, non-parametric identification theory that assumes infinite data. 3) Fig. 1a, the overriding example used throughout the paper is arbitrarily confounded and so it's unclear how valid estimates are obtainable even in an online setting since the effects of all actions are liable to be confounded. Can the authors comment on the identifiability of the approach they describe in Section 2? 4) In line with the above comment, the authors should consider citing Shpitser and Sherman '18 in the paragraph on identifiability, as this work gives a sound and complete for interventions with arbitrary policies in DTR contexts. Additionally, could the authors please comment on the relationship between identifiability of Algorithm 1 estimates and the results posed in SS18? SS18 would seem to indicate that the effects in Alg 1 are not identifiable. 5) "performing a randomized experiment in the actual environment can be extremely costly and undesireable..." and "RL provides a unique opportunity to efficiently learning DTRs due to its nature of balancing exploration and exploitation". Can the authors comment on the difference between a randomized experiment and "exploration" in human subjects research? It is arguably the case that the latter is a glorified (but not ethically distinct) version of the former. 6) In equation mid/bottom of page 3 (unnumbered), where is U? The choice of policy is made in order to maximize a counterfactual value (e.g. Y or the next time step S) and that outcome is thus confounded by U. 7) After thm 2 "The maximum in Eq 3 is achieved with the second best policy" -- can the authors please provide intuition why this is the case? 8) The bound given here seems really poor. For instance, the authors consider a 12-month study for treatment of alcohol abuse. In this case it might be reasonable to consider that there are monthly (12) visits and at each visit the patient could be in one of 3 states (asymptomatic, mild withdrawal symptoms/adverse effects of alcoholism, major withdrawal symptoms/adverse effects of alcoholism). Thus |S| is 3^12. In the simple case, we are just concerned with a single binary treatment so |X| is 2. Realistically other treatments would be considered. If they are related to alcoholism, this exponentially increases |X|, and if they are unrelated, this increases the complexity entailed by the unobserved confounders (see other comments). Even for 1 run of the algorithm, this entails a massive regret lower bound. The authors should consider adding _realistic_ assumptions in order to make this bound more useful. 9) "...which means that UC-DTR is near-optimal provided with only the domains of state S and actions X" -- to be more explicit about point made above, the confounding here due to latent variables could be arbitrarily bad. The theorems in section 2 (thm 1-3) make no assumption or reference to the presence of confounding. Can the authors explain how confounding plays a role in these estimates and these theoretical results? The actions taken are not randomized (policies are assumed deterministic) so past data is effectively "observed" from the standpoint of a decision being made "now". 10) What is the assumed structure of confounding for Sec. 3? Is it Fig 1b? 11) Doesn't causal consistency make Lemma 1 trivial? I.e. the two sides are equal under consistency. 12) gamma(s1) doesn't seem adequately defined in Thm 5. Shouldn't this also specify somehow that x_0 is an undefined quantity? Gamma isn't formally defined to be either a function of 2 variables or of 1. 13) For Cor 1: Y is assumed to be binary, as stated earlier so it seems the bounds simplify to (number in [0,1]) <= counterfactual of interest <= 1 + (number in [0,1]). Why is the upper bound useful if using 1 is at least as tight? 14) In experiments: "The probabilities of the joint distribution (...) are drawn randomly over [0,1]" <- what distribution is used here? 15) In experiments, can the authors please provide justification for using age as the unobserved confounder? Age is almost always observed in healthcare settings. Additionally, age is univariate and arguably gaussian or uniform. This does not pose a very complicated unobserved confounder and it somewhat diminishes the realism of the experiment absent further justification. The authors should consider adding more complicated confounders. 17) Theorem 6 seems to be a DTR version of the non-identifiability theorem posed in Shpitser and Pearl 06 -- ie it's a completeness result on non-identifiability. Such a DTR theorem was proven in Shpitser and Sherman 18. Can the authors comment on the differences between these results?

Reviewer 3



(Originality) Although the use of partial identification bounds for policy learning is not new (Kallus and Zhou 2018; the authors should cite this), yet its use in online decision making is innovative. In particular, the proposed approach to combine partial identification bounds and upper confidence bounds is elegant. (Quality) The proposed method is well developed and the theoretical analysis is thorough. For the empirical evaluation, I think the experiment description is quite terse (especially for cancer treatment) and it may be hard to reproduce the results according to the descriptions alone. The appendix adds some details but making it more complete and clear can be very helpful (e.g., introduction of the dataset, definition of the treatments, how to introduce confounding, etc.). Moreover, in the experiment, I suggest the authors also evaluate the Causal UC-DTR algorithm with $M_t^c$ replaced by confidence bounds learnt from the observational data (as opposed to the bounds from the observation data that incorporate both partial identification and confidence bound). This would show the benefit of taking the confoundedness of the observational data into consideration.

[Author Response · NeurIPS 2019]

We thank the reviewers for their thoughtful feedback. We believe that a few misreadings of our work made some of the
evaluations overly harsh and would ask reviewers to reconsider our paper in the light of clarifications provided below.

**1. Why Online learning? (R2)** It is acknowledged in the literature (Chakraborty and Murphy, 2014; Chakraborty
and Moodie, 2013) that most of learning methods in DTRs focus on the batch (offline) settings, and "development of
statistically sound estimation and inference techniques for" the online reinforcement learning (RL) in DTRs "is an
important research direction." We answers this open problem by proposing the first online RL algorithm in DTRs with
provably theoretical guarantees. The applications of online RL in health care are motivated by the increasing "use
of sophisticated mobile devices" which enables continuous monitoring and intervention on the fly (Chakraborty and
Murphy, 2014). For example, a physician could prescribe a set of safe treatments and use online methods to find the
optimal combination for a patient with chronic conditions. Broadly speaking, when the combination of the observational
data (e.g., $P(\boldsymbol{v})$) and causal knowledge (causal diagram $G$) does not ensure the identifiability of the causal effects
($E_{\boldsymbol{\pi}}[Y]$) in DTRs, the state of art methods would suggest running randomized clinical trials (RCTs) such as SMART
(Murphy, 2003, 2005), and solve for the optimal policy using the standard offline methods (e.g., Q-learning). As the
reviewer (R2) suggested, running online experiments in the real environment could be dangerous and expensive. Our
results are the first adaptive randomization algorithm that could identify the optimal policy in DTRs while achieving
near-optimal bounds on the cost of experimentation (regret). For more discussions on adaptive randomization and
online RL, see (Gittins, 1979; Rosenberger and Lachin, 2015).

**2. High-dimensional State-Action Space (R1, R2)** For experimental studies (e.g., RCTs) in DTRs, issues of sample
complexity could arise when the state-action space $|\boldsymbol{\mathcal{S}}||\boldsymbol{\mathcal{X}}|$ is high-dimensional. It was believed in the DTR literature that
adaptive randomization procedures (e.g., online RL) could circumvent this issue (Cheung, Chakraborty, and Davidson,
2015). Our analysis reveals that this is not the case. In particular, we present first results (Thms. 1-3) analyzing the
information complexity of experimental studies in DTRs. We show that a regret bound of $\Omega(\sqrt{|\boldsymbol{\mathcal{S}}||\boldsymbol{\mathcal{X}}|T})$ is inevitable
for any randomization procedure, regardless of how sophisticated it might be. This suggests that we should explore
other methods to improve the learning convergence. For example, one could exploit the parametric assumptions of the
underlying functions (e.g., linear). Our approach takes another route and tries to leverage the abundant, observational
data that are available prior to the experimental studies. Specifically, we consider the non-identifiable settings where
system dynamics (causal effects $P_{\bar{\boldsymbol{x}}_k}(s_{k+1}|\bar{\boldsymbol{s}}_k)$ and $E_{\bar{\boldsymbol{x}}_K}[Y|\bar{\boldsymbol{s}}_K]$) could not be uniquely determined due to unobserved
confounding in canonical DTR models defined in Def. 1 (e.g., Fig. 1(a)). We derive informative bounds about the
system dynamics in DTRs from the confounded observational data and incorporate the derived bounds into the online
algorithm in an elegant way. We show that this novel approach combining both the online RL and offline bounding
could significantly improve the learning performance of online learners for a large family of DTR instances.

**3. Identification Conditions (R2)** Our online algorithms (Sec. 2.1) are developed under the conditions of sequential
back-door in DTRs (e.g., Fig. 1(b)); while bounding results in Sec. 3.1 are applicable to DTRs with arbitrary unobserved
confounding (Fig. 1(a)). Reviewer 2 (R2) seems to be somewhat confused with our identification conditions, and
might mistook Fig. 1(a) as the basis of causal assumptions used for online RL methods in Sec. 2.1. As R2 suggested,
randomized experiments and online RL are similar in nature (Bareinboim, Forney, and Pearl, 2015). Since each
candidate policy $\boldsymbol{\pi}$ does not take $\boldsymbol{U}$ as input, the sequential backdoor holds in the samples collected by the online RL
algorithm in Alg. 1. Causal quantities $P_{\bar{\boldsymbol{x}}_k}(s_{k+1}|\bar{\boldsymbol{s}}_k)$ and $E_{\bar{\boldsymbol{x}}_K}[Y|\bar{\boldsymbol{s}}_K]$ are thus immediately identifiable, and estimable
from experimental data. For instance, in Fig. 1(b), $S_1, S_2$ block all back-door paths from $X_1, X_2$ to $Y$. The causal effect
$E_{x_1,x_2}[Y|s_1,s_2]$ could be estimated as $E[Y|x_1,x_2,s_1,s_2]$. Given these clarifications, we would like to respectfully
ask R2 to re-evaluate our online RL algorithm (Alg. 1) and theoretical regret analysis (Thms. 1-3).

**R2:** (1) Regarding the factorization of distribution $P_{\boldsymbol{\pi}}(\bar{\boldsymbol{x}}_K, \bar{\boldsymbol{s}}_K, y)$ (below Line 113, Page 3), the exogenous $\boldsymbol{U}$
are subsumed in the product of causal quantities $P_{\bar{\boldsymbol{x}}_k}(s_{k+1}|\bar{\boldsymbol{s}}_k)$ and $E_{\bar{\boldsymbol{x}}_K}[Y|\bar{\boldsymbol{s}}_K]$. Specifically, we average $\boldsymbol{U}$ over
distribution $P(\boldsymbol{u})$, and factorize the resultant causal effect $P_{\bar{\boldsymbol{x}}_K}(y, \bar{\boldsymbol{s}}_K)$ following the basic definition of conditional
probabilities and exclusion restrictions. (2) Bounds in Thm. 2 is a decreasing function relative to $\Delta_{\boldsymbol{\pi}}$ and is maximized
when $\Delta_{\boldsymbol{\pi}}$ is the smallest, i.e., $\boldsymbol{\pi}$ is the second best policy. (3) The consistency axiom suggests that counterfactual
probabilities $P(\bar{\boldsymbol{s}}_{k+1_{\bar{\boldsymbol{x}}_k}}|\bar{\boldsymbol{x}}_k) = P(\bar{\boldsymbol{s}}_{k+1}|\bar{\boldsymbol{x}}_k)$. However, Lem. 1 is concerned with gap between causal quantities
$P_{\bar{\boldsymbol{x}}_k}(\bar{\boldsymbol{s}}_{k+1}) - P_{\bar{\boldsymbol{x}}_k}(\bar{\boldsymbol{s}}_k)$, which generally do not equate to the bound $P(\bar{\boldsymbol{s}}_{k+1}, \bar{\boldsymbol{x}}_k) - P(\bar{\boldsymbol{s}}_k, \bar{\boldsymbol{x}}_k)$. (4) $\Gamma(s_1)$ is well defined
since we define $\bar{X}_k$ as an empty set when $k < 1$. $\Gamma$ is a function over state-action space for all horizon $k = 1, \ldots, K$.
(5) In Corol. 1, the upper bound must be no larger than 1 since $E[Y|\bar{\boldsymbol{s}}_K, \bar{\boldsymbol{x}}_K] \leq 1$. (6) Thm. 6 is *stronger* than standard
non-identifiability proof since for the constructed DTRs $M_1, M_2$, not only their transitional probabilities $P_{\bar{\boldsymbol{x}}_k}(s_{k+1}|\bar{\boldsymbol{s}}_k)$
have to be different, but they also need to *match exactly* the lower and upper bounds in Thm. 5. The bounds in Thm. 5
are thus tight given the confounded observational data. To the best of our knowledge, we are not aware of any other
tightness result regarding DTRs in the literature.

**R1, R3:** We really appreciate the reviewers for the helpful suggestions and references. We will incorporate these
changes in the camera-ready version of the paper if accepted.

[Meta-Review · NeurIPS 2019]

In this paper, the authors provide a method for incorporating observational data (possibly subject to unobserved confounding) to improve the performance of policy learning in online settings (crucial theorems are 5,7 and 8). After a period of discussion, the reviewers came to a consensus that this paper merits publication in NeurIPS, and will contribute to the RL literature by giving a principled method of incorporating observational data, even if confounded.